# GRASP: Graph Reasoning via Agentic Solving and Probing of LLMs

Xiaojun Guo [* 1 2]  Mingxue Tian [* 3]  Chenheng Zhang [1]  Xiaohan Wang [2]  Jiajun Chai [2]  Guojun Yin [2]  Wei Lin [2]
Yifei Wang [4]  Yisen Wang [1]

## Abstract

Integrating graph knowledge into Large Language Models (LLMs) via passive representation faces critical bottlenecks: limited context windows, unreliable numerical computation, and structural hallucinations. To solve this, we propose **GRASP** (**G**raph **R**easoning via **A**gentic **S**olving and **P**robing), shifting the paradigm from passive ingestion to proactive agentic exploration. By interleaving Neighbor Retrieval for on-demand probing with Code Interpreter as a deterministic solver, GRASP enables LLMs to autonomously navigate and compute over complex topologies. We employ a staged reinforcement learning strategy (GRPO) that transitions from visible tuning to a structure-blind environment, forcing the agent to develop genuine topological awareness. Evaluated on multi-domain graph reasoning benchmarks, our 4B model achieves a 53.06% average performance boost, surpassing SOTA baselines like DeepSeek-V3.2 and successfully generalizing to unseen tasks, with high potential for tackling sampling on million-node graphs and solving Hard-level LeetCode graph problems. Our implementation is open-sourced at https://github.com/PKU-ML/GRASP, with models hosted on Huggingface collection PKU-ML/grasp.

## 1. Introduction

Graphs are the foundational language for modeling complex systems, from social networks (Pei et al., 2020) and molecular structures (Borgwardt et al., 2005) to massive knowledge bases (Hu et al., 2020). Traditionally, integrating graph knowledge into Large Language Models (LLMs) has followed a passive representation paradigm. The graph is typically treated as a static input—a fixed adjacency matrix or a flattened text description (Guo et al., 2025b; Luo et al., 2025; Fatemi et al., 2023; Liu et al., 2025b). This passive approach faces several critical bottlenecks: constrained by the LLM's context window capacity, suffering from unreliable numerical computations on complex topologies, and remaining prone to structural hallucinations. Consequently, these models struggle with intricate graph algorithmic problems or scaling to vast networks in the real world.

In this paper, we propose a fundamental paradigm shift: moving from passive ingestion to proactive agentic exploration. Specifically, we introduce **GRASP** that incentivizes **G**raph **R**easoning via **A**gentic **S**olving and **P**robing of LLMs. Inspired by how humans perceive complex scenes, GRASP treats graph understanding as a self-driven process, utilizing a specialized toolkit that functions as its *eyes* for probing and *hands* for solving:

- *Neighbor Retrieval*: This tool enables the model to perform local, on-demand probing of the topology. Like human focal perception, Neighbor Retrieval allows the agent to fetch neighborhood information only when the reasoning path requires it, effectively overcoming the memory constraints of a fixed context window.

- *Code Interpreter*: This tool provides a deterministic environment for symbolic solver, offloading complex topological computations to reliable execution. In a closed-loop feedback cycle, the agent receives real-time traces of Code Interpreter, allowing it to self-correct and debug logic in-context.

By interleaving these tools, the model transitions from a passive observer limited by fixed prompts to a proactive explorer. In this loop, Neighbor Retrieval provides the structural probing evidence, while Code Interpreter acts as a deterministic solver, allowing the agent to selectively query and process information in evolving reasoning states.

To cultivate this proactive agency, we employ a reinforcement learning framework based on Group Relative Policy Optimization (GRPO) (Shao et al., 2024). The RL training incentivizes the model to discover optimal exploration

---

[*]Equal contribution  [1]State Key Lab of General Artificial Intelligence, School of Intelligence Science and Technology, Peking University  [2]Meituan  [3]School of Automation and Intelligent Sensing, Shanghai Jiao Tong University  [4]Amazon AGI SF Lab. Correspondence to: Yisen Wang <yisen.wang@pku.edu.cn>.

*Proceedings of the 43ʳᵈ International Conference on Machine Learning*, Seoul, South Korea. PMLR 306, 2026. Copyright 2026 by the author(s).

strategies by comparing different tool-use trajectories. We involve a critical staged transition: begins by learning tool syntax on visible graphs, and moves to a structure-blind setting where the underlying topologies are masked. In this environment, the agent is forced to develop topological awareness: it can no longer rely on lucky guesses or semantic patterns, but must instead use tools to systematically map out an unknown environment. This ensures that the model's reasoning is grounded in the actual structural evidence it has retrieved, rather than the internal biases of the LLM.

Our evaluations across diverse benchmarks—spanning real-world networks to complex algorithmic tasks—demonstrate that GRASP significantly improves base model performance by an average of 53.06%. Notably, our 4B model surpasses state-of-the-art baselines, including DeepSeek-V3.2 (Liu et al., 2025a) and the reasoning-heavy LongCat-Flash (Team et al., 2025), also showing zero-shot generalization to real-world domains and public graph-theoretic benchmarks. Furthermore, the synergy of GRASP's probing and solving tools present potentials on highly challenging tasks, such as large-scale subgraph sampling on million-node graphs and solving Hard-Level LeetCode graph problems.

The core of our contribution lies in three key innovations:

- We shift from passive graph ingestion to proactive agentic exploration. Our GRASP framework realizes this by framing graph reasoning as a self-driven, tool-embedded process of probing and solving.

- We propose a staged GRPO strategy: transitioning the agent from visible tool-tuning to structure-blind environments, forcing it beyond mimicry toward true topological awareness.

- GRASP empowers a 4B model to beat SOTA LLMs (*e.g.*, DeepSeek-v3.2) on reasoning benchmarks, generalize to unseen tasks, and tackle challenges from million-node graph sampling to Hard LeetCode.

## 2. Related Works

**LLMs Meet Graph Learning.** LLM-graph integration follows two paths: GNN-LLM hybrids and pure LLMs. Hybrids leverage LLMs for feature enhancement (He et al., 2024a; Yu et al., 2025), embedding alignment (Chen et al., 2024b; Wang et al., 2024a), or structural augmentation (Kong et al., 2025), but are limited by GNN-centric inputs. Pure LLM approaches frame graph tasks as language modeling (Wang et al., 2025b), investigating linearization strategies like encoding (Fatemi et al., 2023), modality (Das et al., 2024), and ordering (Chu et al., 2025). To enhance performance, recent works employ supervised fine-tuning (Luo et al., 2024) or preference alignment (Chen et al., 2024a) on graph-specific instruction sets. Despite their success,

these models often suffer from enforced memorization and overfit to specific dataset patterns rather than generalizable topological reasoning.

**RL-driven Graph Reasoning.** Drawing inspiration from the success of Reinforcement Learning from Verifiable Rewards (RLVR) in mathematical and coding domains—most notably exemplified by DeepSeek-R1 (Guo et al., 2025a)—recent studies have begun to treat graph structures as logical environments for RL exploration. G1 (Guo et al., 2025b) first applies this by training models on synthetic graph-theoretic problems with deterministic rewards. Graph-R1 (Luo et al., 2025) reformulates graph tasks as multi-step textual reasoning processes, using task-specific rethink templates to allow the model to self-correct during inference. Further extending this to temporal settings, Ding et al. (2025) utilizes RL to navigate the increased complexity of link forecasting in dynamic graphs.

**Agentic and Tool-Augmented Graph Reasoning.** Parallel research enhances LLMs via agentic frameworks or tool-invocation to mitigate hallucinations, falling into three primary methodologies: modular decomposition, which partitions graph problems into manageable sub-tasks for specialized agents (Hu et al., 2024; Li et al., 2024; Wang et al., 2025a; Yang et al., 2025c); API and algorithm invocation, where models retrieve and execute functions from predefined knowledge bases (Yuan et al., 2025; Zhang, 2023; Wang et al., 2024b); and code-centric paradigms, which solve tasks via executable program generation refined by compiler feedback (Cai et al., 2024; Zhang et al., 2024; Finkelshtein et al., 2025). However, these often rely on manually curated knowledge bases, struggling with out-of-scope tasks and true internalization of structural properties. Conversely, our framework bypasses predefined knowledge bases, employing agentic RL to simultaneously optimize tool proficiency and intrinsic topological understanding.

## 3. Integrated Toolset for Graph Reasoning

We equip GRASP with a dual-capability environment: Neighbor Retrieval for local topology probing and a Code Interpreter for deterministic logic execution. Interleaving these tools enables the agent to transition from passive reception to autonomous exploration, adaptively coordinating structural retrieval and analytical synthesis.

### 3.1. Neighbor Retrieval: Dynamic Receptive Field

For the receptive field of a graph, standard prompting (Fatemi et al., 2023; Wang et al., 2025b; Li et al., 2025) rely on a static context paradigm, where a pre-defined $k$-hop ego-graph is flattened into the query. This approach can cause neighborhood noise (larger $k$) or structural signal starvation (smaller $k$). As demonstrated in Table 1, there is

*Table 1.* Test accuracy (%) of Qwen3-4B-Thinking-2507 on node classification benchmarks.

| Dataset | 0-hop | 1-hop | 2-hop | 3-hop |
|---------|-------|-------|-------|-------|
| Cora | **63.00** | 62.00 | 58.50 | 47.50 |
| PubMed | 70.00 | **72.50** | 46.00 | 27.00 |
| Products | 69.00 | 71.50 | **76.00** | 66.00 |

no universal $k$: Cora peaks with 0-hop (target node only), while Products requires 2-hop context. This necessitates a dynamic approach to balance context density and relevance.

Facing a similar challenge, Graph Neural Networks (GNNs) evolve from fixed-layer aggregations to more expressive, adaptive architectures (Kipf, 2016; Wu et al., 2019). For example, Polynomial GNNs (Chien et al., 2020; He et al., 2021; 2022) adaptively learn coefficients $\theta_k$ to dictate the importance of each $k$-hop expansion. This framework allows the model to mathematically decide which structural scales are most pertinent to the task.

While LLMs cannot directly learn numerical propagation parameters $\theta$, they can leverage semantic reasoning to assess information sufficiency. To achieve this, we equip the LLM with a *Neighbor Retrieval Tool*, shifting the burden of flexibility from a learnable parameter to an agentic decision. If the initial signal is weak, the LLM invokes the tool to expand its receptive field—increasing $k$ on the fly. Mirroring a polynomial filter, the agent discriminates among retrieved neighbors to isolate task-pertinent nodes. This iterative process identifies the minimum sufficient subgraph for a confident decision, replacing fixed-depth aggregation with instance-wise exploration.

### 3.2. Code Interpreter: Closed-Loop Symbolic Execution

Beyond neighbor retrieval, graph tasks often require complex algorithmic reasoning, such as computing minimum spanning trees. Probabilistic LLMs often struggle with the deterministic precision these algorithms demand, leading to path hallucination, where models lose track of visited vertices in long chains. Moreover, simulating deep recursion textually is token-prohibitive, often exhausting context windows before convergence (Guo et al., 2025b).

To resolve these limitations, code execution serves as a robust alternative that ensures mathematical precision and reduces token overhead. However, existing code-based methods (Cai et al., 2024; Zhang et al., 2024) typically implement this via an open-loop pipeline: the LLM generates programs in isolation, which are then executed by an external interpreter to produce static outputs. This paradigm is fundamentally one-way: if the generated code contains errors, the model has no opportunity to intervene and often requires humans to restart the prompt.

In contrast, our tool operates within a closed-loop *Reason-Act-Observe-Correct* cycle (Yao et al., 2022; Madaan et al., 2023). By integrating the interpreter directly into the agent's iterative reasoning steps, the LLM receives instantaneous execution feedback, including runtime tracebacks and intermediate variable states. This creates a self-correction mechanism: if a generated algorithm fails or returns an illogical result, the agent can analyze the error in-context and emit a revised version. This iterative refinement allows the model to abstract complex topological logic into a function call while maintaining the flexibility to debug in real-time.

## 4. Tasks and Experimental Benchmarks

To validate our framework, we evaluate on multi-domain benchmarks covering node/edge-level inference and semantic question answering. To challenge the agent's algorithmic grounding, we introduce RealErdős, a novel benchmark that transforms abstract graph-theoretic problems into narrative-driven, real-world inquiries.

### 4.1. Multi-Domain Graph Reasoning Benchmarks

**Structural and Semantic Graph Reasoning.** We evaluate on established benchmarks with coupled node attributes and topology. These include node classification on co-citation (PubMed (Qian et al., 2023), Arxiv (Hu et al., 2020)), co-purchase (Products (Hu et al., 2020)), and Wikipedia (WikiCS (Mernyei & Cangea, 2020)) networks, alongside link prediction on knowledge graphs (fb15k237, wn18rr (Liu et al., 2024)). While full-graph processing is infeasible at scale, their inherent locality allows accurate inference via strategic neighborhood acquisition.

**Graph-Centric Question Answering.** We incorporate reasoning tasks requiring topology grasp, including TSGBench (Yang et al., 2025b) for scene graph understanding and ExplaGraphs (He et al., 2024b) for commonsense-augmented argument assessment. These tasks require synthesizing natural language reasoning with relational constraints.

**Graph Algorithmic Reasoning.** To evaluate deterministic algorithmic logic, we utilize Erdős (Guo et al., 2025b), encompassing 50 graph-theoretic problems of complexities varying from linear-time to NP-hard. This tests the agent's ability to bypass the traditional struggles of probabilistic token prediction in algorithmic domains.

### 4.2. RealErdős: Bridging Graph Theory and Practical Application

While synthetic graph theoretic benchmarks such as Erdős provide a controlled test environment, they often fail to capture the semantic ambiguity and noisy inquiries of real-world scenarios. We introduce RealErdős, a novel benchmark designed to evaluate how agentic LLMs identify ab-

stract graph-theoretic patterns within human narratives and perform structural grounding.

**Task Composition.** We utilize nine fundamental graph-theoretic problems as structural backbones, each contextualized into five real-world scenarios across diverse domains. For example, bipartite maximum matching is framed as cloud resource allocation, and common neighbor detection as social network interest discovery. This yields 45 unique tasks, preventing reliance on narrow, scenario-specific heuristics; see Appendix B for details.

**Construction and Validation.** The development of Real-Erdős follows a four-stage pipeline: 1) Topology Generation: generating diverse structures via Erdős–Rényi (ERDdS & R&wi, 1959) and Barabási–Albert (Barabási & Albert, 1999) models, ensuring the benchmark covers varying graph distributions. 2) Symbolic Ground Truth: computing deterministic ground truths via the NetworkX library (Hagberg et al., 2008). 3) Semantic Mapping: employing GPT-4o (Hurst et al., 2024) to translate graphs into natural narratives that obscure underlying mathematics. 4) Dual-Verification: implementing a self-correction filter where GPT-4o first attempts to solve the task. Failure cases are manually audited to ensure the semantic ambiguity does not corrupt the original mathematical logic. This pipeline maintains mathematical truth while maximizing narrative authenticity.

# 5. GRASP Framework

In this section, we introduce the proposed GRASP framework, including the tool implementation, problem formulation, and training strategy designs. Detailed system prompt and tool configurations are presented in Appendix A.

## 5.1. Tool Implementation

**Neighbor Retrieval Tool.** To enable active exploration of the local topology, we implement an atomic retrieval tool capable of fetching up to a $k$-hop neighboring information of a target node $u$, where $k$ and $u$ are input parameters for calling the tool function. Given the massive scale of datasets such as Arxiv, we incorporate a cache mechanism to mitigate the latency associated with repeated disk I/O. This ensures that graphs are loaded into memory on demand and persisted across reasoning steps, thereby minimizing disk I/O latency. Furthermore, to maintain the integrity of our evaluation, we implement a data leakage guard: if a retrieved neighbor belongs to the test set, its textual metadata is dynamically masked and replaced with a "Missing" token, preventing possible label or feature contamination.

**Code Interpreter Tool.** The Code Interpreter is designed to transform the LLM's heuristic intent into deterministic output. We execute programs within an isolated sub-process using a *sandboxed-exec* paradigm. This ensures that invalid

memory accesses in graph algorithms do not interfere with the primary agent process. To mitigate the risks associated with LLM-generated code, we implement a restricted execution environment that blocks sensitive system calls, such as file-system writes and network access. Moreover, we limit each execution within a predefined timeout threshold to avoid infinite loops or excessive resource consumption. For the result extraction, rather than relying on standard output streams (which can be noisy), we require the model to assign results to a global result dictionary. This allows for precise, programmatic extraction of symbolic answers.

## 5.2. Staged Agentic Reinforcement Learning

**Trajectory Formulation.** Let $q \sim P(Q)$ represent the input query. A trajectory $\tau$ generated by the policy $\pi_\theta$ is defined as a joint sequence of reasoning traces, environmental actions, and observations:

$$\tau = \{h_0, a_1, \omega_1, h_1, a_2, \omega_2, \ldots, a_T, \omega_T, h_T\}$$

where $h_t \in \mathcal{H}$ denotes the reasoning trace at step $t$; $a_t \in \mathcal{A}$ denotes an external action (e.g., a $k$-hop neighbor query or a Python script execution); $\omega_t \in \Omega$ denotes the environment observation returned by the tools. The probability of a trajectory $\tau$ and the terminal answer $\hat{y}$ under policy $\pi_\theta$ is decomposed as follows:

$$\pi_\theta(\tau, \hat{y}|q) =$$
$$P(h_0|q) \prod_{t=1}^{T} \Big[ P(a_t|q, \tau_{<t}, h_{t-1}) \cdot P(h_t|q, \tau_{<t}, \omega_t) \Big] P(\hat{y}|q, \tau). \tag{1}$$

**Reward Attribution.** For each training query, we sample a group of $G$ trajectories $\{\tau_1, \ldots, \tau_G\}$ from the current policy $\pi_\theta$. We define $\mathcal{E}(\tau, T)$ as an extraction function that retrieves the content associated with block type $T \in$ {think, tool_call, answer}. Following Guo et al. (2025b), the reward $R(\tau_i, s)$ for each trajectory $\tau_i$ is computed based on the terminal answer $\hat{y}_i = \mathcal{E}(\tau_i, \text{answer})$ relative to the ground truth $s$:

$$R(\tau_i, s) = \begin{cases} 1, & \text{if exact\_match}(\hat{y}_i, s) \\ |s \cap \hat{y}_i|/|s \cup \hat{y}_i|, & \text{if } s \text{ is a set} \\ \text{Verifier}(\hat{y}_i, s), & \text{if } s \text{ is not unique} \\ 0, & \text{otherwise,} \end{cases}$$

For unordered sets, the Jaccard Index provides a dense signal for imperfect solutions. For problems with non-unique correct solutions (*e.g.*, shortest paths), we implement algorithmic verifiers to validate the proposed path against the graph's constraints. While we focus on outcome-based rewards in our primary experiments, we discuss the auxiliary tool-use rewards in Section 7.2.

**Policy Updates.** We employ Group Relative Policy Optimization (GRPO) (Shao et al., 2024) for RL training. Unlike standard actor-critic methods, GRPO computes the advantage $A_i$ by normalizing the reward of each trajectory against the group mean:

$$A_i = \frac{R(\tau_i, s) - \text{mean}(\{R(\tau_j, s)\}_{j=1}^G)}{\text{std}(\{R(\tau_j, s)\}_{j=1}^G)}$$

The policy $\pi_\theta$ is updated by maximizing the clipped surrogate objective:

$$\mathcal{J}_{\text{GRPO}}(\theta) = \mathbb{E}_{q, \{\tau_i\}} \frac{1}{G} \sum_{i=1}^G \Big( \min \Big( \frac{\pi_\theta(\tau_i|q)}{\pi_{\theta_{\text{old}}}(\tau_i|q)} A_i,$$

$$\text{clip} \Big( \frac{\pi_\theta(\tau_i|q)}{\pi_{\theta_{\text{old}}}(\tau_i|q)}, 1 - \epsilon, 1 + \epsilon \Big) A_i \Big) - \beta \mathbb{D}_{KL}(\pi_\theta || \pi_{\text{ref}}) \Big),$$

where $\epsilon$ controls the clipping range of the probability ratios. KL divergence from the reference policy $\pi_{\text{ref}}$ prevents catastrophic forgetting and mode collapse. By comparing diverse reasoning traces and tool calls within groups, GRPO penalizes trajectories with faulty logic or inefficient tool use, while rewarding those that successfully ground the reasoning in the graph environment.

**Two-stage RL Training.** While standard benchmarks provide a foundation for tool use, they often invoke a single tool under shallow interactions. More importantly, through mapping task descriptions to classic algorithmic templates, models can succeed without grounding their logic in the specific graph topology. To move beyond this semantic pattern-mimicry, we implement a staged complexity scaling, forcing the emergence of an interleaved reasoning-retrieval-execution loop.

- *Stage I: Tool Familiarization.* Training begins on established benchmarks with full graph descriptions. The agent learns the operational syntax of the Neighbor Retrieval and Code Interpreter tools in isolation.

- *Stage II: Proactive Discovery.* Upon reward convergence, we introduce a blink setting, where the agent receives only a Graph ID to solve a graph-theoretic problem. Withholding the topology renders the task unsolvable via zero-shot code generation.

The transition to the blind setting forces the agent to develop active exploration policies, identifying high-utility nodes instead of relying on algorithmic templates. Since the topology is hidden, the agent must anchor its code in retrieved structural signals, eliminating tool-hacking or lucky guesses. This increases interaction depth from approximately 5 to 60 turns and enables the model to handle unscripted tasks.

# 6. Experiments

## 6.1. Performance on Graph Reasoning

### 6.1.1. SETTINGS AND BASELINES

We utilize Qwen3-4B-Thinking-2507 (Yang et al., 2025a) as our base model, selected for its pretrained agentic ability. We follow the backbone's interaction format, *i.e.*, `<think></think>-<tool_call></tool_call>-<answer></answer>`. We curate a training corpus of 10,410 samples from the benchmarks detailed in Section 4, while maintaining a strictly held-out set for evaluation. We evaluate two primary versions of our agent: **GRASP-Base**, which represents the model after Stage I (Syntactic Familiarization), and **GRASP**, the final model resulting from the full curriculum. The training parameters are provided in Appendix A.3.

We assess our framework against a diverse hierarchy of competitive baselines: 1) *Frontier Generalist LLMs*. We include GPT-4o (Hurst et al., 2024), LongCat-Flash-Thinking-2601(Team et al., 2025), and DeepSeek-V3.2 (Liu et al., 2025a), alongside high-capacity open-source models like the 30B-parameter Qwen3-Thinking and Qwen3-Coder (Yang et al., 2025a). 2) *Tool-Augmented LLMs*. We equip the above generalist models with the two tools proposed in Section 3 to isolate the specific impact of tool synergy. 3) *Domain-Specific LLMs*. We also compare against specialized graph-reasoning methods G1-3B (Guo et al., 2025b), Graph-R1 (Luo et al., 2025), GraphWiz (Chen et al., 2024a), GraphAgent (Yang et al., 2025c), and GraphTool (Wang et al., 2024b). We also consider a No-Tool GRPO ablation model **GRPO-Pure** trained on identical data but restricted to internal reasoning without external tool access.

### 6.1.2. RESULTS

**Comparison with Generalist LLMs.** As shown in Table 2, GRASP demonstrates a substantial leap in performance, achieving an overall average accuracy of 83.85%, outperforming the 4B base model (30.79%) by over 53% and surpassing frontier models like DeepSeek-V3.2 (68.90%) and GPT-4o (47.46%). The comparison between GRASP-Base (Stage I) and GRASP (Stage I & II) validates our staged training strategy: GRASP-Base provides a solid foundation, while full GRASP delivers the boost for advanced reasoning (*e.g.*, 88.57% vs. 86.22% on RealErdős), with its potential emerging on large-scale graph sampling (Section 8).

**Comparison with Tool-Augmented LLMs.** Despite sharing the same tools, the base model lags far behind GRASP (65.68% vs. 83.85%). Our GRASP-4B also surpasses DeepSeek-V3.2 (671B), the best tool-augmented baseline, by 7.80% (83.85% vs. 76.05%). This confirms that tool access alone is insufficient, even for strong backbones, underscoring our training method's effectiveness. As shown

*Table 2.* Test accuracy (%) on diverse downstream graph benchmarks.

| Category | Model | Node Classification | | | | Link Prediction | | Graph QA | | Graph Theory | | *Average* |
|---|---|---|---|---|---|---|---|---|---|---|---|---|
| | | Arxiv | PubMed | Products | WikiCS | fb15k237 | wn18rr | TSG-Bench | ExplaGraphs | Erdős | RealErdős | |
| **Generalist** | Qwen3-4B-Thinking (**base model**) | 51.00 | 25.00 | 21.00 | 29.00 | 16.00 | 13.00 | 62.00 | 45.00 | 38.80 | 7.11 | 30.79 |
| | GPT-4o | 52.00 | 43.00 | 72.00 | 24.00 | 52.00 | 24.00 | 72.00 | 77.00 | 40.60 | 18.07 | 47.46 |
| | LongCat-Flash-Thinking-2601 | 64.00 | 71.00 | 73.00 | 81.00 | 60.00 | 14.00 | 83.00 | 90.00 | 73.40 | 51.20 | 66.06 |
| | DeepsSeek-V3.2 | 65.00 | 47.00 | 70.00 | 79.00 | 65.00 | 26.00 | 88.00 | **99.00** | 83.60 | 66.44 | 68.90 |
| | Qwen3-30B-A3B-Thinking | 59.00 | 70.00 | 52.00 | 62.00 | 58.00 | 26.00 | 72.00 | 4.00 | 30.40 | 12.00 | 44.54 |
| | Qwen3-Coder-30B-A3B | 34.00 | 35.00 | 51.00 | 45.00 | 43.00 | 26.00 | 39.00 | 0.00 | 19.20 | 3.22 | 39.69 |
| **Tool-Augmented** | Qwen3-4B-Thinking (**base model**) | 62.00 | 55.00 | 69.00 | 81.00 | 69.00 | 54.00 | 82.00 | 91.00 | 64.00 | 29.85 | 65.68 |
| | GPT-4o | 58.00 | 42.00 | 69.00 | 71.00 | 65.00 | 35.00 | 82.00 | 95.00 | 76.00 | 59.39 | 52.59 |
| | LongCat-Flash-Thinking-2601 | 65.00 | 55.00 | 71.00 | 82.00 | 62.00 | **79.00** | 84.00 | 97.00 | 77.00 | 52.52 | 70.74 |
| | DeepSeek-v3.2 | 70.00 | 52.00 | 73.00 | 83.00 | **87.00** | 78.00 | **86.00** | 96.00 | 80.00 | 80.00 | 76.05 |
| **Domain-Specific** | G1 | 23.00 | 55.00 | 44.00 | 51.00 | 13.00 | 13.00 | 65.00 | 77.00 | 52.80 | 9.66 | 39.04 |
| | Graph-R1 | 32.00 | 59.00 | 56.00 | 31.00 | 41.00 | 43.00 | 78.00 | 89.00 | 35.20 | 16.59 | 48.08 |
| | GraphWiz | 1.00 | 1.00 | 0.00 | 2.00 | 0.00 | 0.00 | 0.00 | 0.00 | 12.40 | 1.87 | 5.00 |
| | GraphAgent | 26.00 | 50.00 | 52.00 | 52.00 | 23.00 | 18.00 | 56.00 | 78.00 | 1.20 | 3.64 | 24.90 |
| | GraphTool | 22.00 | 61.00 | 50.00 | 51.00 | 14.00 | 20.00 | 38.00 | 71.00 | 15.40 | 5.27 | 32.51 |
| | GRPO-Pure | 60.00 | 82.00 | 74.00 | 83.00 | 69.00 | 43.00 | 80.00 | 56.00 | 53.20 | 40.67 | 64.08 |
| **Ours** | **GRASP-Base-4B** | 69.00 | **91.00** | **78.00** | **88.00** | 86.00 | 68.00 | 85.00 | 95.00 | 89.40 | 86.22 | 83.56 |
| | **GRASP-4B** | **73.00** | 90.00 | 77.00 | **88.00** | 82.00 | 67.00 | 85.00 | 97.00 | **91.00** | **88.57** | **83.85** |

in Appendix D, GRASP's average interaction turns grow from 3 to 15, indicating deeper problem-space exploration. In Appendix F, we also provide an code snippet example within GRASP's closed-loop correction mechanism.

**Comparison with Domain-Specific LLMs.** GRASP outperforms recent graph-reasoning baselines by nearly 35% absolute over the second-best Graph-R1 (48.08%). Our model also surpasses the agentic counterparts (*e.g.*, GraphTool vs. GRASP: 32.51% vs. 83.85%), which typically depend on predefined knowledge bases and result in limited generalization. Moreover, the ablation (GRASP vs. GRPO-Pure: 83.85% vs. 64.08%) reveals the limits of internalizing graph knowledge solely in weights.

**Time Efficiency Comparison.** We report the detailed efficiency metrics in Table 3. For inference efficiency, GRASP significantly reduces total inference time (*e.g.*, 5.48s vs. 13.73s on PubMed) by offloading complex reasoning to external tools. This substantially cuts token consumption (1372 vs. 3667 on PubMed), accelerating the process as LLM latency is dominated by autoregressive decoding. For tool overhead, thanks to our caching and lightweight sandbox, tool-call overhead is minimal (*e.g.*, 0.50s out of 5.48s on PubMed). This efficiency gain, coupled with GRASP's superior accuracy, demonstrates that strategic tool usage offers both effectiveness and speed.

*Table 3.* Efficiency metrics averaged per sample. Format: Inference time (s) / Token consumption / Tool-call time (s).

| Model | PubMed | fb15k237 | Erdős |
|---|---|---|---|
| Qwen3-4B-Thinking | 13.73/3667.18/- | 22.26/5853.13/- | 31.81/8150.06/- |
| **GRASP** | 5.48/1372.24/ 0.50 | 10.19/2573.47/0.22 | 9.51/2520.58/0.002 |

## 6.2. Zero-shot Transferability to Unseen Graph Reasoning Benchmarks

**Zero-shot Real-World Graph Reasoning.** We evaluate our models' zero-shot performance on real-world benchmarks: Cora for citation network classification (Yang et al., 2016), ml1m for movie recommendation (Fey & Lenssen, 2019), and SGDS for scene graph description (Yang et al., 2025b). These test whether the agent can adapt its tool-use strategy to unfamiliar graph semantics. As shown in Table 4, GRASP maintains a strong lead with an average of 64.50% and outperforms specialized baselines like GraphTool (45.83%), highlighting its consistent zero-shot advantage.

*Table 4.* Test accuracy (%) on unseen real-world graph tasks.

| Model | Cora | ml1m | SGDS | *Average* |
|---|---|---|---|---|
| Qwen3-4B-Thinking | 49.00 | **36.00** | 20.50 | 35.16 |
| G1 | 35.50 | 23.50 | 1.50 | 20.17 |
| GraphWiz | 0.00 | 18.00 | 0.00 | 6.00 |
| GraphAgent | 52.50 | 25.50 | 2.00 | 26.66 |
| GraphTool | 55.50 | 32.00 | 50.00 | 45.83 |
| GRPO-Pure | 59.50 | 33.00 | 45.50 | 46.00 |
| **GRASP-Base** | **65.00** | 31.00 | 96.50 | 64.16 |
| **GRASP** | 64.00 | 31.00 | **98.50** | **64.50** |

**Zero-shot Graph Theoretic Reasoning.** We evaluate zero-shot transferability on widely recognized graph-theoretic reasoning benchmarks: GraphArena (Tang et al., 2024), GraphWiz (Chen et al., 2024a), NLGraph (Wang et al., 2023), and GraphInstruct (Luo et al., 2024). These datasets introduce distribution shifts, varying graph sizes, encoding mechanisms, and novel algorithmic tasks. As detailed in Table 5, GRASP achieves consistently high accuracy (*e.g.*,

75.65% on GraphArena and 86.36% on GraphWiz), maintaining a substantial margin over specialized baselines like G1. This cross-benchmark success confirms our method generalizes across formats, fostering autonomous graph exploration and algorithmic synthesis.

*Table 5.* Test accuracy (%) on public graph theoretic benchmarks. Detailed task performances are presented in Appendix C.

| Model | GraphArena | GraphWiz | NLGraph | GraphInstruct |
|---|---|---|---|---|
| Qwen3-4B-Thinking | 50.84 | 44.92 | 40.90 | 52.57 |
| G1 | 24.80 | 48.67 | 53.60 | 36.24 |
| GRPO-Pure | 68.80 | 55.00 | 65.70 | 63.86 |
| **GRASP-Base** | 73.95 | 82.75 | 85.00 | **85.90** |
| **GRASP** | **75.65** | **86.36** | **87.10** | 85.14 |

### 6.3. Comparison with Graph Neural Networks

Graph Neural Networks (GNNs) excel on specific, seen distributions but rely on dataset-specific numerical filters. In contrast, GRASP navigates unseen domains via natural language and strategic tool use. We compare against classical GNNs: GCN (Kipf, 2016), GAT (Veličković et al., 2017), GraphSAGE (Hamilton et al., 2017), and their relational variants R-GCN and R-GAT (Schlichtkrull et al., 2018).

As shown in Table 6 and Table 7, GRASP outperforms classical GNNs across diverse benchmarks, underperforming only on wn18rr, where the purely structural path patterns are particularly conducive to GNNs. Notably, on large-scale arxiv benchmark, GRASP surpasses the best GNN by 12% (73.00% vs 61.00%). We further compare the transferability in Table 8. GCN only achieves 44.50% on Cora even with finetuning, while GRASP achieves 64.00% in a completely zero-shot setting. This confirms that GRASP learns generalized topological awareness rather than just fitting dataset-specific spectral filters.

*Table 6.* Test accuracy (%) on node classification tasks.

| Model | arxiv | pubmed | products | wikics |
|---|---|---|---|---|
| GCN | 58.00 | **90.00** | 70.00 | 82.00 |
| GAT | 61.00 | 89.00 | 72.00 | 84.00 |
| GraphSAGE | 59.00 | 87.00 | 64.00 | 84.00 |
| **GRASP** | 73.00 | 90.00 | 77.00 | 88.00 |

*Table 7.* Test accuracy (%) on link prediction tasks.

| Model | fb15k237 | wn18rr |
|---|---|---|
| R-GCN | 63.00 | **75.00** |
| R-GAT | 67.00 | 73.00 |
| **GRASP** | **82.00** | 67.00 |

*Table 8.* Transferability to unseen graph benchmarks.

| Model | cora |
|---|---|
| GCN (Zero-shot) | 15.00 |
| GCN (Finetuned classifier) | 44.50 |
| **GRASP** (Zero-Shot) | **64.00** |

### 6.4. Analysis of Agent-augmented Utility

We analyze agent utility via the *Tool Success Ratio* (proportion of calls leading to correct solutions). As shown in Figure 1, GRASP (blue) achieves an 82.6% success ratio, far surpassing the base model's 57.1%. This indicates that our RL tuning enables the agent to treat tool outputs as actionable intelligence rather than static noise. Furthermore, tool access resolves 44.1% of queries that previously failed in the no-tool setting, as measured by the *Error Recovery Rate* in Figure 1 (orange). This demonstrates that tool synergy provides the topological grounding necessary to override linguistic hallucinations.

Figure 2 provides a granular assessment of tool-specific performance. The Neighbor Retrieval query distribution (left part) reveals that GRASP performs a significantly more expansive and diverse exploration of the graph topology. We also observe a substantial increase from 72.0% to 92.9% in the Code Interpreter success rate (right part), indicating that GRASP is able to generate executable, algorithmically sound code within the graph environment.

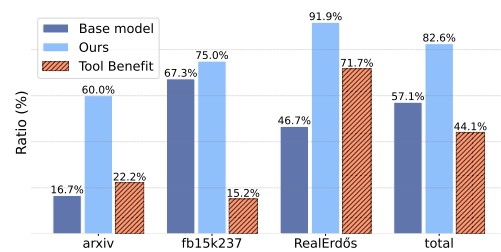

*Figure 1.* Agent-augmented Performance Analysis. Blue bars represent the *Tool Success Ratio*, *i.e.*, percentage of tool calls leading to correct final answers, comparing the base model and GRASP. Orange bars illustrate the *Error Recovery Rate*, the percentage of incorrect no-tool queries successfully resolved by GRASP.

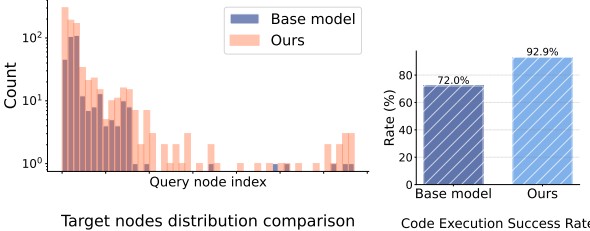

*Figure 2.* Tool details: the distribution of queried nodes for the Neighbor Retrieval Tool and the successful execution rate for the Code Interpreter Tool.

## 7. Ablation Study

### 7.1. Tool Usage

We conduct an ablation study with variants using only Neighbor Retrieval (GRASP-Retrieval) or only the Code In-

terpreter (GRASP-Code). Table 9 shows GRASP-Retrieval excels at semantics-heavy tasks (e.g., Node Classification), while GRASP-Code dominates Graph Theoretic problems. Neither single-tool variant matches the full GRASP framework (68.28%/73.34% vs 83.56%). This confirms that tool complementarity, rather than individual tool strength, is what drives performance.

*Table 9.* Test accuracy (%) for tool usage. We display the average performance of each category. NC: Node Classification; LP: Link Prediction; GQA: Graph QA; GT: Graph Theoretic Problems.

| Tool Usage | NC | LP | GQA | GT | *Average* |
|---|---|---|---|---|---|
| Qwen3-4B-Thinking | 31.50 | 14.50 | 53.50 | 22.95 | 30.61 |
| GRASP-Retrieval | 80.50 | 79.50 | 88.50 | 18.57 | 68.28 |
| GRASP-Code | 73.00 | 52.50 | 87.50 | 80.73 | 73.34 |
| **GRASP-Base** | **81.50** | **77.00** | **90.00** | **87.81** | **83.56** |

### 7.2. Tool Rewards

**Reward Design.** Here, we analyze the effect of additional tool rewards beyond the outcome rewards considered in our primary models. For each tool invocation at step $t$, the agent receives an immediate reward $r_t^{\text{imme}}$:

$$r_t^{\text{imme}} = \begin{cases} r_{\text{f}} + r_{\text{s}} & \text{Correct format, successful execution} \\ r_{\text{f}} + r_{\text{e}} & \text{Correct format, failed execution} \\ 0 & \text{Incorrect format,} \end{cases}$$
$$(2)$$

Here, $r_{\text{f}} = 0.05$ encourages proper tool calling syntax; $r_{\text{s}} = 0.1$ rewards successful tool execution; and $r_{\text{e}} = -0.02$ provides a mild penalty to facilitate learning from errors without gradient collapse.

At the end of an episode, we scale the cumulative immediate reward $R_{\text{total}} = \sum_{t=1}^{T} r_t^{\text{imme}}$ into a final reward $R_{\text{final}}$. For the code interpreter, $R_{\text{final}} = R_{\text{total}} \times \rho_{\text{success}}$, where $\rho_{\text{success}}$ is the execution success ratio. For neighbor retrieval, we introduce an efficiency multiplier: $R_{\text{final}} = R_{\text{total}} \times \rho_{\text{success}} \times \max(\eta_{\text{efficiency}}, 0.5)$, where $\eta_{\text{efficiency}}$ denotes the unique queries over total queries, penalizing redundant looping for diverse exploration. The clamp accommodates necessary re-verification in complex algorithmic tasks.

**Results.** We compare four reward configurations: baseline (no tool rewards), tool-specific ablations (Neighbor Retrieval or Code Interpreter only), and the combined variant. Table 10 reveals that while explicit rewards nearly double tool invocation frequency (25.79% to 45.13%), they slightly degrade task performance. This suggests reward hacking: the model prioritizes dense intermediate rewards from tool use over the final correctness rewards, leading to contextual overload or redundant tool calls for scenarios where tools are unnecessary. Conversely, omitting explicit tool rewards fosters a selective strategy, where the agent utilizes tools

only when indispensable for the final success.

*Table 10.* Test accuracy (%) for tool rewards. We display the average performance of each category. NC: Node Classification; LP: Link Prediction; GQA: Graph QA; GT: Graph Theoretic Problems. Tool Ratio denotes the ratio of tool calls among queries.

| Tool Rewards | NC | LP | GQA | GT | *Average* | *Tool Ratio* |
|---|---|---|---|---|---|---|
| Neighbor Retrieval | 80.50 | 75.00 | 89.00 | 81.18 | 81.42 | 29.22% |
| Code Interpreter | 81.00 | 77.00 | 87.00 | 84.44 | 82.36 | 36.47% |
| Both | 80.75 | **78.00** | 88.50 | 81.87 | 82.28 | 45.13% |
| **No (GRASP-Base)** | **81.50** | 77.00 | **90.00** | **87.81** | **84.07** | 25.79% |

### 7.3. SFT as Cold-Start

**SFT Data Construction.** In this section, we consider an ablation of using supervised finetuned model as the start of RL (Guo et al., 2025a). We construct the SFT training corpus using a rejection sampling pipeline. We first evaluate the base model (Qwen3-Thinking-4B-2507) on our benchmark, isolating only the instances where it fails. For these samples, we use GPT-4o (Hurst et al., 2024) to generate potential solutions with 16 responses per prompt at a high temperature ($T = 1.0$). To maintain the highest data integrity, we discard any response that exhibits incorrect final answers, malformed tool-calling syntax, or conflicting reasoning steps. Finally, we curate a balanced training set with an 8:2 ratio between samples requiring tool invocation and those solvable via pure reasoning.

**Training.** Before the primary reinforcement learning stage, we optimize the policy $\pi_\theta$ using the expert trajectories $\tau^*$ curated through rejection sampling. Using the dataset $\mathcal{D}_{\text{SFT}}$, we minimize the negative log-likelihood of the joint sequence:

$$\mathcal{L}_{\text{SFT}}(\theta) = -\mathbb{E}_{(q,\tau^*)\sim\mathcal{D}_{\text{SFT}}} \left[ \sum_{i=1}^{|\tau^*|} \log \pi_\theta(x_i|q, x_{<i}) \right], \quad (3)$$

where $x_i$ represents each token within $\tau^*$. This stage enforces the causal dependency between the reasoning traces $h_t$, tool actions $a_t$, and observations $\omega_t$ defined in our trajectory formulation.

**Results.** As shown in Table 11, the SFT-Only model provides a strong baseline (68.39%), confirming that our rejection sampling pipeline successfully teaches the model basic tool syntax. However, the most significant performance leap comes from reinforcement learning. While the combined GRASP-SFT achieves the highest average accuracy (85.68%), it only marginally outperforms the pure RL-trained GRASP (84.32%). This narrow 1.36% gap suggests that while SFT can accelerate the learning process and provide a slight edge in complex graph theory (GT), it is not strictly required. The pure RL model demonstrates sufficient capability to discover effective tool-use strategies

from scratch, proving that our RL-training strategy is robust even without expert-supervised data.

*Table 11.* Test accuracy (%) comparing models with or without SFT.

| Model | NC | LP | GQA | GT | *Average* |
|---|---|---|---|---|---|
| Qwen3-4B-Thinking | 31.50 | 14.50 | 53.50 | 22.95 | 30.61 |
| SFT-Only | 61.00 | 56.00 | 84.00 | 72.56 | 68.39 |
| GRASP | **82.00** | 74.50 | **91.00** | 89.78 | 84.32 |
| GRASP-SFT | **82.00** | 80.50 | 86.50 | 93.73 | 85.68 |

## 8. From Atom to Advanced Abilities

### 8.1. Large-scale Subgraph Sampling

We consider a comprehensive task of sampling from the massive ogbn-mag dataset (Hu et al., 2020) (1.9M nodes, 21.1M edges), constrained by specific starting nodes and subgraph sizes. This task is challenging for LLMs: 1) *Topological Exploration*. Since the full graph exceeds context limits, the agent must iteratively invoke Neighbor Retrieval to probe neighbors, requiring understanding the local connectivity to select expansion paths. 2) *Size Monitoring*. As the subgraph grows, the agent requires the Code Interpreter for precise node counting where LLM backbones struggle. 3) *Context Management*. To prevent token overflow in multi-turn interactions, the agent needs to leverage Code Interpreter to offload topology storage to the file system. Overall, this task serves as a rigorous benchmark for tool coordination and topology understanding in a truly large-scale environment.

The results in Figure 3 demonstrate that GRASP achieves near-perfect performance of above 97% on average, surpasses the base model by a large margin even as the target subgraph size increases to 15,000 nodes. In stark contrast, GRASP-Base suffers a catastrophic performance cliff, with accuracy plummeting from 92% to 12% as the sampling depth increases. This divergence highlights the critical value of our two-stage agentic reinforcement learning that enforces transition from shallow pattern mimicry to proactive agentic reasoning. These results show that GRASP is capable of reliable, autonomous exploration in massive real-world environments where static models fail.

### 8.2. Leetcode Graph Problems

Our primary benchmarks focus on atomic algorithmic abilities—applying the Code Interpreter to solve foundational graph theoretic problems (*e.g.*, connectivity or basic traversal). However, real-world reasoning often requires advanced abilities of synthesizing complex, multi-stage logic where the reasoning path is not well-established. To determine if our models can bridge this gap, we evaluate on graph algo-

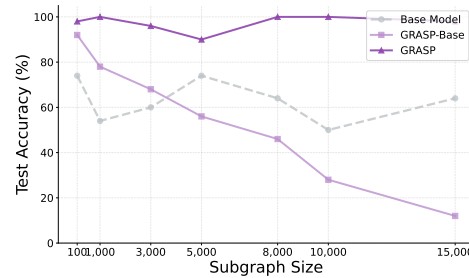

*Figure 3.* Test accuracy (%) on the graph sampling task. For each subgraph size, we perform 50 runs by varying the start node.

rithm problems in Leetcode[1]. Specifically, we evaluate our model on 8 LeetCode graph problems—5 Medium-Level and 3 Hard-Level. Details are shown in Appendix E.

As shown in Figure 4, GRASP achieves an average accuracy of 39.25%, a substantial improvement over the base model's 24.00%. Notably, it outperforms GRASP-Base on Hard-level problems: for instance, 44% vs. 32% on lc2192, and 50% vs. 42% on lc2097. These results underscore the value of our staged agentic reinforcement learning, which helps the model move beyond template-based code generation toward more flexible and adaptive problem-solving.

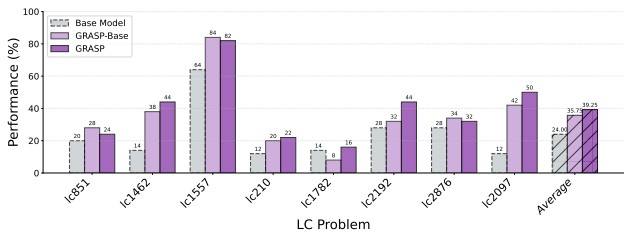

*Figure 4.* Test accuracy (%) on Leetcode graph problems.

## 9. Conclusion

In this paper, we introduce GRASP, a framework that redefines graph reasoning in LLMs by shifting from a passive ingestion paradigm to proactive agentic exploration. By integrating a dual-tool system into a staged RL training strategy via GRPO, we successfully addressed the long-standing bottlenecks of context window limitations, numerical instability, and structural hallucinations in graph-based tasks. GRASP proves that small-scale models, when equipped with the right exploratory tools and reasoning frameworks, can achieve competitive reasoning capabilities with larger systems. Future work will focus on expanding this agentic paradigm to multi-modal graph-text environments, further pushing the boundaries of autonomous graph intelligence.

---

[1]https://leetcode.com/

## Acknowledgements

Yisen Wang is supported by National Natural Science Foundation of China (62376010, 92370129), Beijing Major Science and Technology Project under Contract no. Z251100008425006, Beijing Natural Science Foundation (No. L257007), Beijing Nova Program (20230484344, 20240484642), and State Key Laboratory of General Artificial Intelligence.

## Impact Statement

This paper presents a methodological framework, GRASP, designed to investigate proactive agentic exploration in graph reasoning. By equipping Large Language Models with specialized tools and reinforcement learning, we evaluate their ability to navigate and solve complex topological tasks. We believe its core contribution is methodological, and we have no specific societal impacts to highlight at this stage.

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

# A. GRASP Details

## A.1. System Prompt

---
**System Prompt**

You are a helpful AI assistant capable of using various tools.
Core Principles:
1. Never generate false or fabricated information.
2. Use tools when external information is required.
3. Provide direct answers when no tool is needed.

Response Format Rules:
[Situation 1: Need to think/reason before answering]
`<think>`Your reasoning process here...`</think>`
`<answer>`Final answer based on your reasoning...`</answer>`

[Situation 2: Have enough information to answer directly]
`<answer>`Direct answer here...`</answer>`

[Situation 3: Need to use a tool]
`<think>`Explain why you need to use a tool`</think>`
`<tool_call>`"name": "tool_name", "arguments": "arg_name": "value"`</tool_call>`

Tool calls MUST use this exact JSON format:
{
"name": "tool_name",
"arguments": {
"arg1": "value1",
"arg2": "value2"
}
}

Examples:
{"name": "query_neighbor_information", "arguments": {"node_index": 1, "data_source": "cora", "k": 1}}
{"name": "execute_code", "arguments": {"code_string": "x=3\ny=12\nresult=x+y\n"}}

[Situation 4: After receiving tool response]
`<tool_response>`The tool's response will appear here`</tool_response>`

You now have two options:

Option A: If you need more information from another tool
`<think>`Explain why you need to use another tool`</think>`
`<tool_call>`{"name": "next_tool_name", "arguments": {...}}`</tool_call>`

Option B: If you have sufficient information
`<think>`Analyze all gathered information`</think>`
`<answer>`Final comprehensive answer`</answer>`
(to be continued)

---

---

### System Prompt (Continued)

Important Notes:
- After a tool call, the system will return results within `<tool_response>` tags
- You can think and make additional tool calls after receiving tool responses
- Continue until you have sufficient information to provide a final answer
- Always use the specified tag format
- Keep responses clear and concise

— TOOL CONFIGURATION —
{tool_configuration}

---

## A.2. Tool Configuration

### Neighbor Retrieval Tool

type: "function"
name: "query_neighbor_information"
description: "Retrieves information about the neighbors of a specified node within a graph structure. This function allows you to query up to 'k' hops away from the given node, using a specific dataset name."
parameters:

- node_index: type: "integer", description: "The unique index identifier of the node whose neighbors you want to query."

- dataset_name: type: "string", description: "The dataset name or database from which the node's neighbor information will be retrieved."

- k: type: "integer", description: "The number of hops away from the node to include in the query. A '1-hop' query will return direct neighbors, a '2-hop' will include neighbors of neighbors, etc. NOTE: for data from arxiv, k should be less than 3 to avoid out-of-memory."

required: ["node_index", "dataset_name", "k"]

### Code Interpreter Tool

type: "function"
name: "execute_code"
description: "Execute the given code and return the results. The code will be executed using Python's exec() function in an isolated environment. Requirements: 1. Code must be self-contained and executable. 2. Standard library imports are available, and include all necessary imports at the top. 3. Use variable assignments to store results (not print statements). 4. **MANDATORY**: Assign the final result to a variable named 'result'. This variable will be used to extract the execution result."
parameters:

- code_string: type: "string", description: "The string of Python code to be executed."

required: ["code_string"]

## A.3. Training Details

We provide the training hyperparameters of SFT and RL in Table 12 and Table 13.

*Table 12.* Training configurations of SFT. LR denotes Learning Rate, BSZ denotes Batch Size.

| Setting | LR | Min LR | LR Scheduler | Warmup Ratio | Weight Decay | BSZ | Max-Length | Data Num. | Epoch |
|---------|------|--------|--------------|--------------|--------------|-----|------------|-----------|-------|
| SFT | 1e-5 | 1e-6 | Cosine | 0.05 | 0.1 | 64 | 8192 | 1042 | 3 |

*Table 13.* Training configurations for RL. Ent. denotes the entropy coefficients. KL. denotes the KL loss coefficients.

| Setting | LR | $T$ | Ent. | KL. | BSZ | Max-Length | Steps | Rolllouts | Max turns |
|---------|------|-----|------|-------|-----|------------|-----------------|-----------|----------------|
| RL | 1e-6 | 1.0 | 0.00 | 0.001 | 32 | 8192 | 700 (Base)/ 1000 | 8 | 10 (Base) / 100 |

# B. Details of RealErdős

We provide an overview of RealErdős in Table 14 and a specific example below.

*Table 14.* An overview of tasks in RealErdős.

| Graph Theory | Tasks |
|--------------|-------|
| **Domination Set** | Wireless Network Coverage, Dominating Set in Social Networks, Data Center Power Optimization, Critical Infrastructure Protection, Sensor Network Deployment |
| **Common Neighbor** | Friend Recommendation in Social Networks, Collaborative Filtering in E-commerce, Link Prediction in Knowledge Graphs, Common Supplier Identification, Shared Interests Detection in Social Networks |
| **Bipartite Maximum Matching** | Job Assignment in Organizations, Resource Allocation in Cloud Computing, Supply Chain Partner Matching, Student-Teacher Assignment, Ad Placement Optimization |
| **Max Weight Matching** | Optimal Matching in Auctions, Job Scheduling Optimization, Optimal Network Design, Financial Portfolio Optimization, Cloud Service Load Balancing |
| **Closeness Centrality** | Shortest Path in Road Networks, Crisis Management in Social Media Networks, Optimal Delivery Route Planning, Telecommunication Network Optimization, Urban Mobility Analysis |
| **Shortest Path** | GPS Navigation Systems, Flight Path Optimization, Robotic Path Planning, Warehouse Logistics Optimization, Network Routing in Communication Systems |
| **Global Efficiency** | Power Grid Efficiency, Global Supply Chain Optimization, Transport System Optimization, Energy Distribution Networks, Urban Traffic Management |
| **Maximal Independent Set** | Network Connectivity Optimization, Resource Allocation in Wireless Networks, Optimal Facility Location Problem, Sensor Placement in Environmental Monitoring, Multi-agent Path Planning |
| **Maximum Flow** | Traffic Flow Optimization, Network Bandwidth Allocation, Water Distribution System, Supply Chain Network Optimization, Emergency Response System |

---

**Example of Realerdős: Social Network Influencers**

The marketing team at VibeStream is planning a seed campaign to promote a new feature. Their budget allows them to partner with only a few key influencers, but they need the announcement to appear on the feed of every user in a specific beta group: Uma, Celebrity_B, Frank, Olivia, Follower_A, Community_Leader_1, Xavier, Yara, and Celebrity_A.

The platform's interaction logs show the following social links: Uma frequently interacts with Celebrity_B, Frank, Follower_A, Xavier, and Celebrity_A. Celebrity_B is in a group chat with Follower_A, Community_Leader_1, Xavier, and Yara. Frank is followed by Olivia and Follower_A. Meanwhile, Olivia often reposts content from Community_Leader_1 and Yara. Follower_A is a verified fan of Celebrity_A. Community_Leader_1 is mutually connected with Xavier and Celebrity_A. Finally, Xavier and Celebrity_A are direct collaborators.

**Question:** To ensure that 100% of these users see the campaign through at least one direct connection, what is the smallest number of seed users the team must contact, and who are they?

## C. Detailed Results of Graph-Theoretic Problems

### C.1. Detailed Results for GraphWiz

We provide the performance details of Graphwiz benchmark in Table 15.

*Table 15.* Test accuracy (%) on the GraphWiz benchmark.

| Model | cycle | connect | bipartite | topology | shortest | triangle | flow | hamilton | subgraph |
|---|---|---|---|---|---|---|---|---|---|
| Qwen3-4B-Thinking | 72.25 | 45.75 | 57.50 | 27.25 | 40.50 | 23.00 | 51.50 | 41.25 | 45.25 |
| G1-3B | 93.00 | 61.25 | 64.00 | 11.75 | 13.50 | 17.75 | 39.00 | 93.75 | 44.00 |
| GRPO-Pure | 92.00 | 97.00 | 51.00 | 27.00 | 38.50 | 23.75 | 47.00 | 53.75 | 65.00 |
| **GRASP-Base** (Ours) | 97.50 | 97.50 | 97.50 | 92.00 | 96.75 | 29.50 | 94.00 | 70.50 | 69.50 |
| **GRASP** (Ours) | 96.25 | 100.00 | 93.75 | 88.25 | 95.50 | 49.00 | 99.50 | 84.00 | 71.00 |

### C.2. Detailed Results for NLGraph

We provide the performance details of NLGraph benchmark in Table 16.

*Table 16.* Test accuracy (%) on the NLGraph benchmark.

| Model | gnn | connectivity | cycle | flow | hamilton | matching | shortest | topology |
|---|---|---|---|---|---|---|---|---|
| Qwen3-4B-Thinking | 82.05 | 44.20 | 41.88 | 65.52 | 67.24 | 0.00 | 62.50 | 11.85 |
| G1-3B | 0.00 | 91.11 | 64.40 | 10.34 | 8.62 | 5.95 | 26.56 | 31.11 |
| GRPO-Pure | 76.92 | 90.84 | 82.20 | 63.79 | 56.90 | 0.00 | 75.00 | 11.11 |
| **GRASP-Base** (Ours) | 84.62 | 99.46 | 93.72 | 100.00 | 74.14 | 10.71 | 90.62 | 74.81 |
| **GRASP** (Ours) | 79.49 | 99.19 | 97.91 | 98.28 | 75.86 | 20.24 | 78.12 | 86.67 |

### C.3. Detailed Results for GraphInstruct

We provide the performance details of GraphInstruct benchmark in Table 17.

*Table 17.* Test accuracy (%) on the GraphInstruct benchmark.

| Model | bfs | dfs | mst | bipartite | clustering | common | component | connectivity | cycle | degree | diameter | edge | euler | hamilton | jaccard | flow | neighbor | pagerank | predecessor | shortest | topological |
|---|---|---|---|---|---|---|---|---|---|---|---|---|---|---|---|---|---|---|---|---|---|
| Qwen3-4B-Thinking | 33 | 19 | 60 | 11 | 38 | 89 | 73 | 63 | 65 | 75 | 80 | 17 | 15 | 74 | 26 | 77 | 77 | 3 | 69 | 78 | 62 |
| G1-3B | 7 | 1 | 2 | 8 | 49 | 78 | 10 | 88 | 88 | 84 | 50 | 95 | 0 | 12 | 35 | 8 | 77 | 8 | 16 | 43 | 2 |
| GRPO-Pure | 72 | 40 | 62 | 21 | 52 | 90 | 86 | 91 | 91 | 63 | 80 | 77 | 10 | 90 | 29 | 64 | 72 | 12 | 78 | 84 | 77 |
| **GRASP-Base** (Ours) | 98 | 99 | 96 | 35 | 95 | 100 | 99 | 100 | 94 | 99 | 100 | 100 | 39 | 82 | 66 | 99 | 100 | 9 | 100 | 100 | 94 |
| **GRASP** (Ours) | 97 | 99 | 92 | 30 | 97 | 99 | 96 | 98 | 94 | 100 | 99 | 100 | 52 | 81 | 59 | 97 | 99 | 7 | 100 | 93 | 99 |

### C.4. Detailed Results for GraphArena

We provide the performance details of GraphArena benchmark in Table 18 and Table 19.

*Table 18.* Test accuracy (%) on the **easy** problems from the GraphArena benchmark.

| Model | Connected | Diameter | Distance | Neighbor | GED | TSP | MCP | MCS | MIS | MVC |
|---|---|---|---|---|---|---|---|---|---|---|
| Qwen3-4B-Thinking | 52.0 | 28.0 | 36.0 | 60.0 | 44.0 | 67.0 | 98.0 | 87.0 | 98.0 | 88.0 |
| G1-3B | 52.0 | 42.0 | 47.0 | 89.0 | 30.0 | 17.0 | 27.0 | 20.0 | 32.0 | 22.0 |
| GRPO-Pure | 89.0 | 54.0 | 87.0 | 100.0 | 59.0 | 90.0 | 98.0 | 91.0 | 99.0 | 93.0 |
| **GRASP-Base** (Ours) | 100.0 | 94.0 | 100.0 | 100.0 | 47.0 | 89.0 | 91.0 | 72.0 | 94.0 | 93.0 |
| **GRASP** (Ours) | 100.0 | 99.0 | 98.0 | 100.0 | 48.0 | 78.0 | 99.0 | 67.0 | 96.0 | 95.0 |

*Table 19.* Test accuracy (%) on the **hard** problems from the GraphArena benchmark.

| Model | Connected | Diameter | Distance | Neighbor | GED | TSP | MCP | MCS | MIS | MVC |
|---|---|---|---|---|---|---|---|---|---|---|
| Qwen3-4B-Thinking | 60.0 | 26.0 | 42.0 | 45.0 | 3.0 | 2.0 | 59.0 | 5.0 | 69.0 | 48.0 |
| G1-3B | 19.0 | 12.0 | 25.0 | 51.0 | 3.0 | 0.0 | 0.0 | 0.0 | 1.0 | 7.0 |
| GRPO-Pure | 95.0 | 20.0 | 88.0 | 99.0 | 10.0 | 6.0 | 65.0 | 7.0 | 68.0 | 58.0 |
| **GRASP-Base** (Ours) | 98.0 | 87.0 | 93.0 | 99.0 | 6.0 | 21.0 | 73.0 | 3.0 | 56.0 | 63.0 |
| **GRASP** (Ours) | 95.0 | 95.0 | 89.0 | 98.0 | 7.0 | 10.0 | 81.0 | 2.0 | 79.0 | 77.0 |

# D. Training Analysis of GRASP and GRASP-Base

We present the training curves of GRASP and GRASP-Base, including the rewards, entropy loss, response length, and mean number of turns in Figure 5. GRASP significantly increases the number of interaction turns (d and h) compared to GRASP-Base, reflecting an enhancement in interaction complexity and deeper engagement with the problem space. This indicates that GRASP is capable of more thorough reasoning and exploration, which is crucial for solving complex graph tasks. The response length (c, g) shows similar values to GRASP-Base. The stable rewards (a, e) and entropy values (b, f) further highlight the stability of GRASP's training process. These improvements collectively demonstrate GRASP's ability to handle complex problems with greater depth, flexibility, and stability.

# E. Leetcode Graph Problems

We present the eight Leetcode graph problems evaluated in our paper below:

---

**Leetcode2097**

**Valid Arrangement of Pairs [Hard]**
You are given a 0-indexed 2D integer array pairs where $\text{pairs}[i] = [\text{start}_i, \text{end}_i]$. An arrangement of pairs is valid if for every index i where $1 <= i < \text{pairs.length}$, we have $\text{end}_{i-1} == \text{start}_i$.

Return any valid arrangement of pairs.

Note: The inputs will be generated such that there exists a valid arrangement of pairs.

---

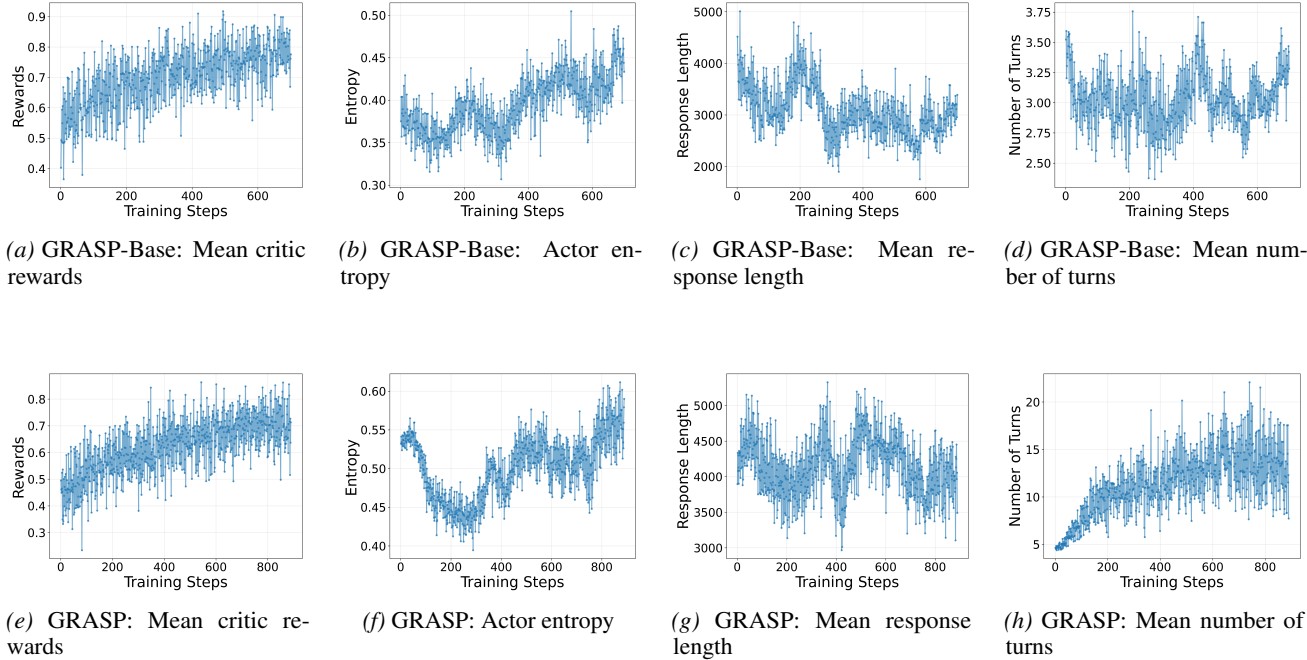

*(a)* GRASP-Base: Mean critic rewards

*(b)* GRASP-Base: Actor entropy

*(c)* GRASP-Base: Mean response length

*(d)* GRASP-Base: Mean number of turns

*(e)* GRASP: Mean critic rewards

*(f)* GRASP: Actor entropy

*(g)* GRASP: Mean response length

*(h)* GRASP: Mean number of turns

*Figure 5.* Training metrics for GRASP and GRASP-Base. Each panel presents one of the four metrics: (a) Critic rewards, (b) Actor entropy, (c) Response length, and (d) Number of turns. The first row (GRASP-Base) shows the performance of the GRASP-Base model, while the second row (GRASP) presents the performance of the GRASP model.

---

**Leetcode851**

**Loud and Rich [Middle]**
There is a group of n people labeled from 0 to n - 1 where each person has a different amount of money and a different level of quietness.

You are given an array richer where richer[i] = [ai, bi] indicates that ai has more money than bi and an integer array quiet where quiet[i] is the quietness of the ith person. All the given data in richer are logically correct (i.e., the data will not lead you to a situation where x is richer than y and y is richer than x at the same time).

Return an integer array answer where answer[x] = y if y is the least quiet person (that is, the person y with the smallest value of quiet[y]) among all people who definitely have equal to or more money than the person x.

---

**Leetcode1462**

**Course Schedule IV [Middle]**
There are a total of numCourses courses you have to take, labeled from 0 to numCourses - 1. You are given an array prerequisites where prerequisites[i] = [ai, bi] indicates that you must take course ai first if you want to take course bi.

For example, the pair [0, 1] indicates that you have to take course 0 before you can take course 1.

Prerequisites can also be indirect. If course a is a prerequisite of course b, and course b is a prerequisite of course c, then course a is a prerequisite of course c.

You are also given an array queries where queries[j] = [uj, vj]. For the jth query, you should answer whether course uj is a prerequisite of course vj or not.

Return a boolean array answer, where answer[j] is the answer to the jth query.

## Leetcode1557

**Minimum Number of Vertices to Reach All Nodes [Middle]**
Given a directed acyclic graph, with n vertices numbered from 0 to n-1, and an array edges where edges[i] = [fromi, toi] represents a directed edge from node fromi to node toi.

Find the smallest set of vertices from which all nodes in the graph are reachable.

It's guaranteed that a unique solution exists.

Notice that you can return the vertices in any order.

## Leetcode210

**Course Schedule II [Middle]**
There are a total of numCourses courses you have to take, labeled from 0 to numCourses - 1. You are given an array prerequisites where prerequisites[i] = [ai, bi] indicates that you must take course bi first if you want to take course ai.

For example, the pair [0, 1], indicates that to take course 0 you have to first take course 1.

Return the ordering of courses you should take to finish all courses. If there are many valid answers, return any of them. If it is impossible to finish all courses, return an empty array.

## Leetcode1782

**Count Pairs Of Nodes [Hard]**
You are given an undirected graph defined by an integer n, the number of nodes, and a 2D integer array edges, the edges in the graph, where edges[i] = [ui, vi] indicates that there is an undirected edge between ui and vi. You are also given an integer array queries.

Let incident(a, b) be defined as the number of edges that are connected to either node a or b.

The answer to the jth query is the number of pairs of nodes (a, b) that satisfy both of the following conditions:

a < b
incident(a, b) > queries[j]

Return an array answers such that answers.length == queries.length and answers[j] is the answer of the jth query.

Note that there can be multiple edges between the same two nodes.

## Leetcode2192

**All Ancestors of a Node in a Directed Acyclic Graph [Middle]**
You are given a positive integer n representing the number of nodes of a Directed Acyclic Graph (DAG). The nodes are numbered from 0 to n - 1 (inclusive).

You are also given a 2D integer array edges, where edges[i] = [fromi, toi] denotes that there is a unidirectional edge from fromi to toi in the graph.

Return a list answer, where answer[i] is the list of ancestors of the ith node, sorted in ascending order.

A node u is an ancestor of another node v if u can reach v via a set of edges.

**Leetcode2876**

**Count Visited Nodes in a Directed Graphh [Hard]**

There is a directed graph consisting of n nodes numbered from 0 to n - 1 and n directed edges.

You are given a 0-indexed array edges where edges[i] indicates that there is an edge from node i to node edges[i].

Consider the following process on the graph:

You start from a node x and keep visiting other nodes through edges until you reach a node that you have already visited before on this same process.

Return an array answer where answer[i] is the number of different nodes that you will visit if you perform the process starting from node i.

# F. Response Example of GRASP

---

**Example of Code Snippets Generated by GRASP**

**Prompt**: The task is to determine the maximal matching in a bipartite graph. The input graph is guaranteed to be a bipartite graph. Here is an undirected graph containing nodes from 1 to 26. The edges are: (1, 12), (1, 19), [omitted for space limit], (11, 21), (11, 24). Question: What is the bipartite maximal matching of the bipartite graph? You need to format your answer as a list of edges in ascending dictionary order, e.g., [(u1, v1), (u2, v2), ..., (un, vn)]. If necessary, you can provide Python code as a string argument to the code execution tool.

**GRASP Response**: `<think>`Okay, let's see. I need to find the maximal matching in this bipartite graph. Hmm, first, what's a maximal matching? .... Let me check. To find a maximum matching, I can use algorithms like Hopcroft-Karp or DFS-based matching. But since this is a large graph (26 nodes), doing it by hand would be tedious. However, the user expects me to use tools here. Wait, but the tools I have are query_neighbor_information and execute_code. The query_neighbor_information function can get neighbors of a node up to k hops. But how does that help with finding a matching? Alternatively, maybe I can write a Python script using networkx to compute the maximum matching. ... **So perhaps I should use the execute_code tool to run a Python script that constructs the graph and computes the maximum matching.** ... `</think>` `<tool_call>` {"name": "execute_code", "arguments": "code_string": "import networkx as nx \n\n # Create a bipartite graph\n G = nx.Graph()\n\n# Add all nodes from 1 to 26\n for i in range(1, 27):\n G.add_node(i)\n\n# Add all edges\n edges = [(1,12), (1,19), [omitted for space limit], (11,21), (11,24)]\n for u, v in edges:\n G.add_edge(u, v)\n\n # Compute maximum matching\n matching = nx.bipartite.maximum_matching(G)\n\n # Convert to list of edges\n result_edges = [(u, v) for u, v in matching.items()]\n result_edges.sort()\n\n # Output as list\n result_edges"} `</tool_call>`

**Tool Results:** `<tool_response>`{"error": "Internal tool execution error: Variable 'result' not found in executed code"}`</tool_response>`

**GRASP Response**: `<think>`Okay, let's see. The code I ran gave an error: "Variable 'result' not found in executed code". Hmm. Wait, the tool requires that the code must assign the final result to a variable named 'result'. In my code, I have 'result_edges' as the list, but I didn't name it 'result'. That's probably why the tool is complaining. **So I need to adjust the code to make sure the final result is stored in a variable called 'result'.** .... Let's try this code again.`</think>` `<tool_call>` {"name": "execute_code", "arguments": "code_string": "import networkx as nx\n\n G = nx.Graph()\n for i in range(1, 27):\n G.add_node(i)\n\n edges = [(1,12), (1,19), [omitted for space limit, (11,21), (11,24)]\n for u, v in edges:\n G.add_edge(u, v)\n\n matching = nx.bipartite.maximum_matching(G)\n result_edges = [(u, v) for u, v in matching.items()]\n result_edges.sort()\n result = result_edges"} `</tool_call>`

**Tool Results**: `<tool_response>`[(1, 12), (2, 23), (3, 15), (4, 25), (5, 14), (6, 13), (7, 22), (8, 16), (9, 17), (10, 24), (11, 20), (12, 1), (13, 6), (14, 5), (15, 3), (16, 8), (17, 9), (20, 11), (22, 7), (23, 2), (24, 10), (25, 4)]`</tool_response>`

**GRASP Response**: `<think>`Okay, let's see. The tool response gave a list of edges. ... I need to present it as a list of tuples in the specified format. `</think>` `<answer>`[(1, 12), (2, 23), (3, 15), (4, 25), (5, 14), (6, 13), (7, 22), (8, 16), (9, 17), (10, 24), (11, 20), (12, 1), (13, 6), (14, 5), (15, 3), (16, 8), (17, 9), (20, 11), (22, 7), (23, 2), (24, 10), (25, 4)]`</answer>`

