# OpenReview forum: "GRASP: Graph Reasoning via Agentic Solving and Probing of LLMs"
_ICML.cc/2026/Conference — ICML 2026 regular_

### Official Review · Reviewer_GkiP · 2026-03-04

**Soundness:** 3
**Presentation:** 4
**Significance:** 3
**Originality:** 3
**Overall Recommendation:** 4
**Confidence:** 4

**Summary:**

The paper proposes GRASP, a framework that enables LLMs to solve graph reasoning tasks by actively probing graph structure and executing code instead of relying on a fully provided graph prompt. The model interacts with two tools: neighbor retrieval (to explore graph topology) and a code interpreter (to perform algorithmic computation) in a closed-loop reasoning process. Empirical results suggest GRASP improves across multiple graph tasks on a wide range of datasets.

**Compliance With Llm Reviewing Policy:**

Affirmed.

**Final Justification:**

My final recommendation is 4 (borderline accept). My main concerns in the original review were the lack of stronger baselines, missing ablation studies, and limited analysis of complexity/efficiency; based on the rebuttal, these concerns have been sufficiently addressed. While additional qualitative case studies could further strengthen the paper, I do not view them as a core issue that would block a positive recommendation.

I still consider the originality to be moderate rather than outstanding, since the contribution lies more in the integration of existing ideas, the training strategy, and the empirical validation than in introducing a fundamentally new primitive. For that reason, I remain at borderline accept rather than accept. Overall, I view this as a solid and well-executed piece of work that demonstrates that this combination of components can be effective in practice.

**Key Questions For Authors:**

1. What part of the performance gain comes from better exploration policy vs better code synthesis?

2. How sensitive is GRASP to query budget and context limits? Can you give a brief comparison between the token cost to the other baseline?

**Limitations:**

Yes

**Strengths And Weaknesses:**

**Strengths**

1. The experiments' suits are very broad with clear gains. The author has shown extensive experimental evidence on a very large scale of datasets, which is very plausible and convincing.

2. Many engineering details are presented, and the curriculum (2-stage) learning idea is very interesting.

3.  The proposed pipeline is carefully designed and systematically evaluated. The interaction between graph probing and code execution is clearly described, and the methodology is implemented in a clean and modular way, making the approach relatively easy to understand and potentially reproducible.


**Weaknesses**

1. **Limited novelty.** The overall framework appears to combine existing components (tool-augmented LLM reasoning, code execution, neighbor retrieval, and RL-based training).

2. **Baseline coverage is incomplete.** The evaluation primarily compares against neighbor-retrieval style probing. Stronger comparisons would include:
     - **GraphRAG-style retrieval** methods that retrieve subgraphs rather than only local neighborhoods. E.g., G-retriever style [1] or GNN-RAG style [2].
     - **API tool baselines**, where the model invokes graph primitives (e.g., shortest path, connectivity, components) instead of synthesizing full code.
     - **Prompt-only tool usage baselines** (no RL training) to isolate the contribution of the proposed training procedure.

3. **Limited evidence that the model learns meaningful graph reasoning.** It remains unclear whether the model truly learns exploration strategies. Some case study can be presented to clearly show the benefits of RL training, for example:
     - Retrieval sequences
     - Generated code snippets
     - Examples showing how the closed-loop correction mechanism fixes mistakes during execution

4. **Model scale and backbone diversity**
   Experiments are conducted primarily on a single backbone (Qwen-4B). Evaluations on larger models (e.g., 7B+) or alternative backbones would strengthen the generality of the claims.

5. **Additional tool ablations are missing.** The current ablations mostly compare against a “no-tool” variant. More granular studies would clarify the role of each component, such as Neighbor retrieval only vs. code execution only vs. both

[1] He X, Tian Y, Sun Y, et al. G-retriever: Retrieval-augmented generation for textual graph understanding and question answering[J]. Advances in Neural Information Processing Systems, 2024, 37: 132876-132907.

[2] Mavromatis C, Karypis G. Gnn-rag: Graph neural retrieval for large language model reasoning[J]. arXiv preprint arXiv:2405.20139, 2024.

---

> ### Author Rebuttal · Authors · 2026-03-31
>
> **W1.** **Limited novelty.**
>
> **A1.** We respectfully clarify that GRASP is not a simple combination of components, but implements a fundamentally different approach to how LLMs interact with and reason over graph structures:
> * **Active Probing.** Unlike [1-3], which treat graphs as **static, flattened text**, GRASP enables **on-demand probing**. This bypasses the input length constraints and reasoning bottlenecks inherent in exhaustive graph flattening.
> * **Closed-loop Solving.** While prior work uses static code snippets, GRASP is the first to **interleave structural probing with real-time symbolic execution**. This allows the agent to write, execute, and debug code dynamically based on environmental feedback.
> * **Topological Grounding.** Our **two-stage RL** moves beyond semantic pattern mimicry to enforce topological reasoning. This enables handling **million-node graphs** and reasoning depths up to **60 turns**—capabilities beyond traditional static baselines.
>
> ----
>
>  **W2.** **Baseline coverage is incomplete.**
>
> **A2.** We have conducted additional experiments to address these points:
> * **GraphRAG-style Baselines**: We compared GRASP with G-Retriever [1] on its original benchmarks (Table R1). GRASP **notably surpasses** G-Retriever variants on *ExplaGraphs* and *Scene Graphs* (e.g., **92.78% vs. 87.05%**). Even in a **zero-shot** setting on *WebQSP*, GRASP achieves a competitive **68.00%**.
> * **API-tool vs. Full Code Synthesis**: We clarify that our Code Retrieval Tool provides a lightweight sandbox where the agent calls standard packages, rather than synthesizing complex code from scratch.
> * **Prompt-only tool baseline**: As shown in Table R2, despite having tool access, the base model significantly lags behind GRASP (**65.68% vs. 83.85%**). This gap validates the necessarity of our RL training.
> ⠀
> **We will incorporate these additional baseline comparisons and discussions in the revised manuscript.**
>
> **Table R1. Test Performance on [1]’s benchmark.**
> | Model              | **expla graphs** | **scene graphs** | **webqsp**        |
> |-|-|-|-|
> | G-Retriever        | 85.16            | 81.31            | 70.49             |
> | G-Retriever w/LoRA | 87.05            | 86.83            | **73.79**         |
> | **GRASP (Ours)**   | **92.78**        | **91.51**        | 68.00 (zero-shot) |
>
> **Table R2. Test accuracy (%) comparison with Prompt-only tool baselines.**
> |                             | NC    | LP    | GQA   | GT    | Average |
> |-|-|-|-|-|-|
> | Qwen3-4B-Thinking-2507-Tool | 66.75 | 61.50 | 86.50 | 46.92 | 65.68   |
> | **GRASP**                   | 82.00 | 74.50 | 87.50 | 89.78 | 83.85   |
>
> ---
>
> **W3.** **Limited evidence that the model learns meaningful graph reasoning.**
>
> **A3.** We thank the reviewer for this constructive suggestion, which will significantly strengthen our paper. Due to character limits, we cannot present the full reasoning traces here. **We commit to adding these case studies to the revised manuscript** and will include retrieval sequences, code snippets, and error-correction examples. **If space permits, we will also provide these traces later in the rebuttal response.**
>
> ---
>
> **W4.** **Model scale and backbone diversity**
>
> **A4.**  To verify generality, we extend GRASP to Qwen3-8B. Results in **Table R3** reinforce our claims: GRASP-8B significantly outperforms vanilla Qwen3-8B (**61.05% vs 86.64%**), and Stage II improves GRASP-Base by 2.45%. We will add these results in a model scalability subsection.
> **Table R3. Accuracy (%) of 8B models.**
>
> |                   | NC    | LP  | GQA  | GT    | Average |
> |-|-|-|-|-|-|
> | Qwen3-8B          | 63    | 42  | 87   | 50.28 | 61.05   |
> | **GRASP-Base-8B** | 81    | 80  | 91.5 | 87.45 | 84.19   |
> | **GRASP-8B**      | 81.25 | 86  | 92   | 92.71 | 86.64   |
>
> ---
> **W5.** **Additional tool ablations are missing.**
>
> **A5.** We conduct an ablation study with variants using only **Neighbor Retrieval (GRASP-Retrieval)** or only the **Code Interpreter (GRASP-Code)**. **Table R4** shows GRASP-Retrieval excels at semantics-heavy tasks (e.g., Node Classification), while GRASP-Code dominates Graph Theoretic problems. Neither single-tool variant matches the full GRASP framework (**68.28%/73.34% vs 83.56%**). We will add this to the manuscript.
>
> **Table R4. Ablation Study on Tool Components.**
>
> |                       | NC   | LP   | GQA  | GT    | Average |
> |-|-|-|-|-|-|
> | GRASP-Retrieval       | 80.5 | 79.5 | 88.5 | 18.57 | 68.28   |
> | GRASP-Code            | 73.0 | 52.5 | 87.5 | 80.73 | 73.34   |
> | **GRASP-Base** | 81.5 | 77.0 | 90.0 | 87.81 | 83.56   |
>
> ---
>
> **Q1.** What part of the performance gain comes from?
>
> **A6.** Please See **A5**.
>
> ---
> **Q2.** How sensitive is GRASP to query budget and context limits? Can you give a brief comparison between the token cost to the other baseline?
>
> **A7.** For time limits, we are working hard for the first question. For the second question, please kindly refer to Reviewer 3qBr's A2.
>
> ----

---

> > ### Author Rebuttal · Reviewer_GkiP · 2026-04-01
> >
> > The rebuttal addresses my main concerns by adding stronger baseline comparisons, larger-model results, and component ablations. I would encourage the authors to include case studies illustrating the RL-learned exploration patterns, and potentially some discussion of the difficulties encountered during RL training for additional insight. I will raise my score to 4.

---

> > > ### Author Response · Authors · 2026-04-03
> > >
> > > We sincerely thank the reviewer for the positive feedback and for raising the score to Weak Accept. We will incorporate the extended baseline comparisons, results on larger models, and component ablations into the manuscript as promised.
> > >
> > > Below, we provide an example of the code snippets generated by GRASP within our closed-loop correction mechanism. Additional case studies, along with a detailed discussion of the challenges encountered during RL training, will be included in the appendix. Thank you again for your constructive questions, which have greatly helped enhance our paper.
> > >
> > > ---
> > >
> > > **Example of Code Snippets Generated by GRASP**
> > >
> > > `Prompt:` The task is to determine the maximal matching in a bipartite graph. The input graph is guaranteed to be a bipartite graph. Here is an undirected graph containing nodes from 1 to 26. The edges are: (1, 12), (1, 19), [omitted for space limit], (11, 21), (11, 24). Question: What is the bipartite maximal matching of the bipartite graph? You need to format your answer as a list of edges in ascending dictionary order, e.g., [(u1, v1), (u2, v2), ..., (un, vn)]. If necessary, you can provide Python code as a string argument to the code execution tool.
> > >
> > > `GRASP Response:` \<think>Okay, let's see. I need to find the maximal matching in this bipartite graph. Hmm, first, what's a maximal matching? …. Let me check. **To find a maximum matching, I can use algorithms like Hopcroft-Karp or DFS-based matching. But since this is a large graph (26 nodes), doing it by hand would be tedious. However, the user expects me to use tools here.** Wait, but the tools I have are query_neighbor_information and execute_code. The query_neighbor_information function can get neighbors of a node up to k hops. But how does that help with finding a matching? **Alternatively, maybe I can write a Python script using networkx to compute the maximum matching.** … So perhaps I should use the **execute_code tool** to run a Python script that constructs the graph and computes the maximum matching. … \</think> **\<tool_call>** {"name": "execute_code", "arguments": {"code_string": "import networkx as nx\n\n# Create a bipartite graph\nG = nx.Graph()\n\n# Add all nodes from 1 to 26\nfor i in range(1, 27):\n G.add_node(i)\n\n# Add all edges\nedges = [(1,12), (1,19), [omitted for space limit], (11,21), (11,24)]\nfor u, v in edges:\n G.add_edge(u, v)\n\n# Compute maximum matching\nmatching = nx.bipartite.maximum_matching(G)\n\n# Convert to list of edges\nresult_edges = [(u, v) for u, v in matching.items()]\nresult_edges.sort()\n\n# Output as list\nresult_edges"}} **\</tool_call>"**
> > >
> > > `Tool Results:` <\tool_response>**{"error": "Internal tool execution error: Variable 'result' not found in executed code"}**\</tool_response>
> > >
> > > `GRASP Response:` \<think>Okay, let's see. The code I ran gave an error: "Variable 'result' not found in executed code". Hmm. Wait, **the tool requires that the code must assign the final result to a variable named 'result'. In my code, I have 'result_edges' as the list, but I didn't name it 'result'. That's probably why the tool is complaining**. So I need to adjust the code to make sure the final result is stored in a variable called 'result'. ….  Let's try this code again.\</think> **\<tool_call>** {"name": "execute_code", "arguments": {"code_string": "import networkx as nx\n\nG = nx.Graph()\nfor i in range(1, 27):\n G.add_node(i)\n\nedges = [(1,12), (1,19), [omitted for space limit, (11,21), (11,24)]\nfor u, v in edges:\n G.add_edge(u, v)\n\nmatching = nx.bipartite.maximum_matching(G)\nresult_edges = [(u, v) for u, v in matching.items()]\nresult_edges.sort()\nresult = result_edges"}} **\</tool_call>**"
> > >
> > > `Tool Results:` \<tool_response>[(1, 12), (2, 23), (3, 15), (4, 25), (5, 14), (6, 13), (7, 22), (8, 16), (9, 17), (10, 24), (11, 20), (12, 1), (13, 6), (14, 5), (15, 3), (16, 8), (17, 9), (20, 11), (22, 7), (23, 2), (24, 10), (25, 4)]\</tool_response>
> > >
> > > `GRASP Response:` \<think>Okay, let's see. The tool response gave a list of edges. ... I need to present it as a list of tuples in the specified format. \</think> \<answer>[(1, 12), (2, 23), (3, 15), (4, 25), (5, 14), (6, 13), (7, 22), (8, 16), (9, 17), (10, 24), (11, 20), (12, 1), (13, 6), (14, 5), (15, 3), (16, 8), (17, 9), (20, 11), (22, 7), (23, 2), (24, 10), (25, 4)]\</answer>"

---

### Official Review · Reviewer_Ax8o · 2026-03-11

**Soundness:** 3
**Presentation:** 3
**Significance:** 3
**Originality:** 3
**Overall Recommendation:** 4
**Confidence:** 4

**Summary:**

GRASP proposes shifting LLM-based graph reasoning from passive graph ingestion to proactive agentic exploration. The framework equips an LLM with two tools—Neighbor Retrieval for on-demand topology probing and a Code Interpreter for deterministic computation—and trains the agent via a two-stage GRPO reinforcement learning style: Stage I teaches tool syntax on visible graphs, and Stage II forces exploration under a "structure-blind" setting where the topology is masked. Evaluated on node classification, link prediction, graph QA, graph-theoretic problems, and a new benchmark, the 4B-parameter model achieves several improvement over the base model and surpasses existing strong backbone models across multiple tasks.

**Compliance With Llm Reviewing Policy:**

Affirmed.

**Final Justification:**

I appreciate the authors' response. After the rebuttal discussion, as my concerns have been adequately addressed, I maintain my positive rating.

**Key Questions For Authors:**

Please see weaknesses.

**Limitations:**

Please see weaknesses.

**Strengths And Weaknesses:**

**Strengths**
1. This paper proposes a fresh, active approach to graph reasoning. Instead of just feeding fixed graph snippets to a LLM, the agent explores the graph step by step—like a human would. This is well-motivated by showing that static methods fail when the "right" amount of context varies across tasks.
2. In the second training stage, the model only gets a graph ID and must learn to ask for information itself, without seeing the whole graph upfront. This increases how deeply it interacts with the graph and is clearly necessary for handling large graphs.
3. The method is evaluated on several fronts: standard benchmarks, new real-world datasets, classic graph problems, coding challenges, and even zero-shot transfer to unseen setups. This shows it’s not just specifically tuned to one task.

**Weaknesses**
1. How can you effectively distinguish whether the method, trained in a structure-blind environment, has simply learned a mapping function or has truly attained topological awareness?
2. Adding a visualization of the pipeline to the paper would greatly facilitate reader comprehension of the method's details.
3. Some math and definitions are unclear or inconsistent. Key equations mix up what’s generated when, and important parts of the reward system, like the "verifier", aren’t formally defined.
4. In the results of ml1m in Table 3, performance actually gets worse. More discussion would be helpful. On LeetCode-style problems, it still fails more than half of the time (about 60% error rate), but this isn’t further explored as well. Moreover, the "million-node" experiment skips practical metrics like runtime or cost.

---

> ### Author Rebuttal · Authors · 2026-03-31
>
> **Q1.** How can you effectively distinguish whether the method, trained in a structure-blind environment, has simply learned a mapping function or has truly attained topological awareness?
>
> **A1.** We thank the reviewer for this insightful question. We clarify that the **structure-blind environment** in Stage II is specifically designed to **force** the emergence of topological awareness rather than simple mapping.
> * In this environment, the graph structure is entirely removed from the model’s initial context. For tasks like “The shortest path from node 2 to 6” with a *node-set* answer type, the agent starts with **zero structural features**, making it mathematically **impossible** to "guess" the answer via a mapping function.
> * Success is only possible if the agent **actively reconstructs** the topology by reasoning which neighbors to probe and how to assemble the returned local segments.
> * The high accuracy (**RL rewards**) under this blind setting signifies successful proactive structural grounding, representing a topological awareness incentivized by the RL training.
>
> ---
>
> **Q2.**  Adding a visualization of the pipeline to the paper would greatly facilitate reader comprehension of the method's details.
>
> **A2.** We thank the reviewer for this constructive suggestion. We agree that a visual pipeline will significantly enhance the clarity of our framework and will incorporate it in the revised manuscript.
>
> ---
>
> **Q3.** Some math and definitions are unclear or inconsistent. Key equations mix up what’s generated when, and important parts of the reward system, like the "verifier", aren’t formally defined.
>
> **A3.** We sincerely thank the reviewer for pointing out the inconsistencies in our mathematical formulations. We will strictly revise Section 5.2 to ensure technical clarity:
> * **Trajectory Formulation**. To avoid the circularity in our original probability decomposition, we redefine the trajectory $\tau$ as the sequence of reasoning and actions, *i.e.*, $\tau = \{h_0, a_1, \omega_1, h_1, a_2, \omega_2, \dots, a_T, \omega_T, h_T\}$, while treating the final answer $\hat{y}$ as a terminal output conditioned on $\tau$.
> * **Clarified Generative Steps**. We will explicitly separate policy-controlled variables ($h_t, a_t$) from environment-determined observations ($\omega_t$) in Equation (1) to clarify "what is generated when”, *i.e.*, $\pi_\theta(\tau, \hat{y} | q) = P(h_0 | q) \prod_{t=1}^T \Big[ P(a_t | q, \tau_{<t}, h_{t-1}) \cdot P(h_t | q, \tau_{<t}, \omega_t) \Big] P(\hat{y} | q, \tau)$.
> * **Formal Definition of "Verifier”**. We clarify that for tasks with **non-unique** solutions like shortest paths, the Verifier is a deterministic algorithmic check $\mathcal{V}: \mathcal{Y} \times \mathcal{S} \times \mathcal{G} \rightarrow \{0, 1\}$ that validates whether a proposed solution $\hat{y}$ (*e.g.*, a path) satisfies the structural constraints of the graph $\mathcal{G}$.
>
> ----
>
> **Q4.** In Table 3, ml1m performance degrades—more analysis needed. LeetCode error rate exceeds 60% but is unexplored. Million-node experiment lacks runtime or cost metrics.
>
> **A4.** We thank the reviewer for the constructive feedback. We provide further analysis on the dataset performance and “million-node” efficiency as follows:
> * **Analysis of ml1m Performance.** The *ml1m* dataset involves predicting subjective user ratings (1-5) for movies, which is heavily influenced by **subjective variance** (*e.g.*, two users may rate the same movie differently despite similar reasoning). This inherent noise leads to a generally lower accuracy (~30%) across all baselines. While GRASP is marginally lower (31% vs. 33%), our model achieves a **substantial 18.67% absolute improvement in average accuracy** across all unseen tasks (64.50% vs. 45.83%), demonstrating superior overall generalization.
> * **Discussion on LeetCode Challenges.** The selected Medium/Hard level LeetCode tasks are inherently challenging even for humans, with official pass rates of **52.3%** (LC-1462) and **66.5%** (LC-2097). Despite using a relatively small **4B-parameter base model**, GRASP achieves remarkable **relative performance leaps**, *e.g.*, **14.0% → 44.0%** (LC-1462) , **12.0% → 50.0%** (LC-2097).
> * **Efficiency on Million-Node Graphs.** As shown in **Table R1**, sampling a 15,000 subgraph takes around 2.3 minites. The primary latency stems from the backbone LLM’s token generation, while the tool-call overhead is minimal (1.15s).
> We will incorporate these discussions and runtime metrics into the revised manuscript.
>
> **Table R1. Efficiency of GRASP on sampling a 15,000-node subgraph from million-node graph. Metrics are averaged across 50 runs.**
>
> | Inference time (s) | Token consumption | Tool-call time (s) | Tool-call number |
> |-|-|-|-|
> | 138.69             | 32841.36          | 1.15               | 17.58            |
>
> ---
>
> We appreciate your time in reviewing and hope these responses help. Please let us know if there are any other points you wish to discuss.

---

> > ### Author Rebuttal · Reviewer_Ax8o · 2026-04-03
> >
> > Thanks for the clarification. I trust the final version will reflect these updates. I will keep my positive score.

---

> > > ### Author Response · Authors · 2026-04-04
> > >
> > > We thank the reviewer for the positive feedback and thoughtful questions. In the revised manuscript, we will incorperate a visual pipeline, the efficiency of million-node graph sampling, further analysis on certain performance aspects, and revise the formulation in Section5.2 following your suggestions.

---

### Official Review · Reviewer_YarX · 2026-03-13

**Soundness:** 3
**Presentation:** 3
**Significance:** 2
**Originality:** 3
**Overall Recommendation:** 4
**Confidence:** 3

**Summary:**

GRASP reframes LLM-based graph reasoning from passive ingestion to proactive agentic exploration. The authors identify three critical bottlenecks: context windows, numerical errors, structural hallucinations and solves using Neighbor Retrieval for dynamic probing, Code Interpreter for deterministic execution, two-stage RL via GRPO. Stage 1 establishes tool proficiency on visible graphs, Stage 2 enforces topological awareness by withholding topology. Authors introduce RealErdos, bridging abstract algorithms with real-world narratives. Evaluation spans node classification, link prediction, graph QA, algorithmic reasoning across multiple benchmarks including a 1.9M-node subgraph sampling and LeetCode graph problems. 4B GRASP achieves 83.85% accuracy, surpassing frontier models.

**Compliance With Llm Reviewing Policy:**

Affirmed.

**Final Justification:**

I thank the authors for running the experiment I requested and I am glad to see that GRASP  wins by 7.8% against a 600+ Billion model with  the same tools. Thanks for clarifying that the ground truth is deterministic (NetworkX) and not LLM-generated. GNN comparisons are also showing that GRASP is competitive or superior on most tasks, with dramatically better zero-shot transfer. As a result, I will raise my score to 4.

**Key Questions For Authors:**

Could GNN-LLM hybrid (GNN embeddings + GRASP) outperform pure GRASP? Without GNN comparisons, it is impossible to assess whether GRASP's approach is genuinely useful or merely demonstrates that LLMs can emulate what specialized architectures already do well.

**Limitations:**

Yes

**Strengths And Weaknesses:**

Strengths:

- The problem is well-motivated by preventing pattern memorization. Empirical validation shows clear benefits on hard tasks. Tool reward analysis, SFT vs. pure RL, agent utility metrics provide solid grounding. Finding that combined tool rewards degrade performance is insightful.

- The proposed benchmark (RealErdos) is novel bridging abstract algorithms with real-world narratives.


Weaknesses:

- GPT-4o used for both generation and checking, risking circularity. Manual auditing sample sizes and inter-annotator agreement is not reported.

- The 4B model has access to two external tools (Neighbor Retrieval and Code Interpreter), while the frontier LLMs are evaluated zero-shot without tools. This is not a fair comparison of reasoning capability. A fairer comparison would equip GPT-4o with the same Neighbor Retrieval and Code Interpreter tools (via function calling) and compare. GRPO-Pure (64.08%, no tools) is probably the more honest capability comparison.

- The authors position the work in the LLM-for-graphs space but does not compare against GNN baselines or GNN-LLM hybrids. GNNs handle the same problems of node classification, link prediction, often with sub-second inference on graphs GRASP spends 5-60 agentic turns on.

---

> ### Author Rebuttal · Authors · 2026-03-31
>
> We thank Reviewer YarX for recognizing our well-motivated problem, clear benefits on hard tasks, solid analysis, and novel RealErdős. Your insightful questions help us improve the manuscript. Below, we summarize and address your concerns.
>
> ---
>
> **W1.** GPT-4o used for both generation and checking, risking circularity. Manual auditing sample sizes and inter-annotator agreement is not reported.
>
> **A1.** We thank the reviewer for this critical observation, which we address as follows:
> * **Circularity Concerns.** The symbolic ground truth is generated deterministically using the **NetworkX library**. GPT-4o acts strictly as a semantic mapper to obscure the mathematics. To ensure this process does not disturb the underlying logic, we conducted **manual checks** to prevent any question-answer mismatch.
> * **Manual auditing sample sizes.** We present the auditing sample size in **Table R1**.
> * **Inter-Annotator Agreement.** Two human experts (PhD students) independently verified the failure samples. We observed an inter-annotator agreement of **93.00%**. Discrepancies were resolved through discussion until a 100% consensus was reached.
>
> We will incorporate the above discussions into the revised manuscript.
>
> **Table R1. Manual auditing sample sizes for RealErdős.**
> | Domination Set | Common Neighbor | Bipartite Maximum Matching | Weight Matching | Closeness Centrality | Shortest Path | Global Efficiency | Maximal Independent | Maximum Flow |
> |-|-|-|-|-|-|-|-|-|
> | 66             | 56              | 81                         | 98              | 76                   | 51            | 99                | 85                  | 88           |
>
> ---
>
> **W2.** The frontier LLMs are evaluated zero-shot without tools.
>
> **A2.** We evaluate three SOTA LLMs by providing them with the same tools. Summarized in **Table R2**, our **4B GRASP** outperforms **the second best DeepSeek-V3.2 (671B parameters)** **by 7.80% on average** (83.85% vs. 76.05%). The results show that simply having access to tools is insufficient even for strong LLM backbones, highlighting the effectiveness of our training method. We will add the results in the revised version.
>
> **Table R2. Test Accuracy (%) comparing tool-equipping baselines.**
>
> |                             | Node Classification | Link Prediction | Graph QA | Graph Theory | Average   |
> |-|-|-|-|-|-|
> | GPT-4o                      | 60.00               | 50.00           | 88.50   | 67.69           | 52.59     |
> | LongCat-Flash-Thinking-2601 | 68.25               | 70.50           | 90.50   | 64.76           | 70.74     |
> | DeepSeek-V3.2               | 69.50               | 42.00           | 91.00   | 80.00           | 76.05     |
> | GRASP                       | 82.00               | 74.50           | 87.50   | 89.78           | **83.85** |
>
> ---
>
>  **W3. & Q1.** The authors does not compare against GNN baselines or GNN-LLM hybrids.
>
> **A3.** Thanks for this critical question regarding the positioning of GRASP relative to GNNs.
> - **Comparison with GNN Baselines.** As shown in **Tables R3-R4**, GRASP outperforms classical GNNs/relational variants on most benchmarks. Notably, on ogbn-arxiv, GRASP surpasses the best GNN by **12%** (73% vs. 61%).
> - **Zero-shot Transferability.** As shown in **Table R5**, even with a fine-tuned classifier, GCN achieves only **44.5%** on Cora when trained on Arxiv. In contrast, GRASP achieves **64.00%** in a completely **zero-shot** setting.
> - **Comparison with GNN-LLM hybrid baselines.** Given the strict character constraints, we kindly invite you to refer to our response to Reviewer **3qBr’ Q3** for a detailed comparison.
>
> We agree that GNNs are highly efficient for specific distributions. However, our GRASP shifts from **pattern fitting** to **generalized reasoning**, empowering models to navigate unseen domains via natural language and strategic tool use, rather than dataset-specific filters. We will incorporate these GNN/GNN-LLM baselines into the revised manuscript.
>
> **Table R3. Accuracy (%) on node classification.**
>
> |                  | Arxiv     | PubMed    | Products  | WikiCS    |
> |-|-|-|-|-|
> | GCN              | 58.00     | **90.00** | 70.00     | 82.00     |
> | GAT              | *61.00*   | *89.00*   | *72.00*   | *84.00*   |
> | GraphSAGE        | 59.00     | 87.00     | 64.00     | *84.00*   |
> | **GRASP** | **73.00** | **90.00** | **77.00** | **88.00** |
>
> **Table R4. Accuracy (%) on link prediction.**
>
> |                  | fb15k237  | wn18rr    |
> |-|-|-|
> | R-GCN            | 63.00     | **75.00** |
> | R-GAT            | 67.00     | 73.00     |
> | **GRASP** | **82.00** | 67.00     |
>
> **Table R5. Tansferability to unseen benchmarks.**
>
> |                            | Cora  |
> |-|-|
> | GCN (zero-shot)            | 15.00 |
> | GCN (Finetuned classifier) | 44.50 |
> | **GRASP (zero-shot)**      | 64.00 |
> ----
> We appreciate your time in reviewing our work and hope these responses resolve your queries. Please let us know if there are any other points you wish to discuss.

---

> > ### Author Rebuttal · Reviewer_YarX · 2026-04-03
> >
> > I thank the authors for running the experiment I requested and I am glad to see that GRASP  wins by 7.8% against a 600+ Billion model with  the same tools. Thanks for clarifying that the ground truth is deterministic (NetworkX) and not LLM-generated. GNN comparisons are also showing that GRASP is competitive or superior on most tasks, with dramatically better zero-shot transfer. As a result, I will raise my score to 4.

---

> > > ### Author Response · Authors · 2026-04-04
> > >
> > > We sincerely thank the reviewer for the positive feedback and for raising the score to Weak Accept. Your constructive  questions are very helpful in improving the quality of our paper. We will add the manual verification details of RealErdős, the tool-equipped LLM baselines, and the GNN and GNN-Hybrid baselines to the revised manuscript as promised.

---

### Official Review · Reviewer_3qBr · 2026-03-13

**Soundness:** 3
**Presentation:** 3
**Significance:** 3
**Originality:** 2
**Overall Recommendation:** 4
**Confidence:** 3

**Summary:**

This paper proposes GRASP, a graph reasoning framework that combines Neighbor Retrieval and Code Interpreter with a two-stage RL training strategy. Instead of relying only on static graph descriptions, GRASP supports more active graph exploration through tool use. Furthermore, GRASP is evaluated on multiple graph reasoning tasks and further introduces RealErdos, a benchmark that reformulates graph-theoretic problems into narrative and real-world scenarios.

**Compliance With Llm Reviewing Policy:**

Affirmed.

**Final Justification:**

The authors have adequately addressed my main concerns regarding computational efficiency, missing baselines, and novelty. The additional experiments in Table R1, R2, and R3 provide convincing evidence of GRASP's effectiveness and efficiency gains. The rebuttal clarifies that GRASP's contribution lies in the principled integration of active probing, closed-loop solving, and topological grounding rather than individual components in isolation. Overall, I view this as a solid piece of work and will raise my score from 3 to 4 (Weak accept).

**Key Questions For Authors:**

See the above **Weaknesses**.

**Limitations:**

Yes

**Strengths And Weaknesses:**

**Strengths**

1. The motivation that fixed graph prompts are insufficient for complex graph reasoning is reasonable, especially for tasks requiring iterative topology exploration and exact computation.
2. Using neighbor retrieval for topology exploration and code execution for precise computation is a sensible combination, and the two-stage training setup is consistent with this design.
3. The paper includes main benchmark results, tool-use analysis, transfer evaluation, and large-graph experiments, which help show the effectiveness of the proposed GRASP across different settings.

**Weaknesses**

1. Tool use, code execution, and RL-based training for graph reasoning have already been explored in prior work [1-3]. The main contribution seems to be the specific combination of these components, rather than a fundamentally new graph reasoning paradigm.
2. The paper shows gains in effectiveness, but does not provide explicit analysis of runtime, token consumption, tool-call overhead, or RL training cost. This is important because the proposed method relies on multi-turn interaction and two-stage RL, both of which may introduce substantial computational overhead.
3. Although the reported results are strong, the paper does not directly compare against several relevant recent graph reasoning methods, e.g., [1-4]. This makes it difficult to judge whether the reported accuracy gains are truly state-of-the-art.

**Reference**
[1] GraphAgent: Agentic Graph Language Assistant, EMNLP, 2025.
[2] GraphTool-Instruction: Revolutionizing Graph Reasoning in LLMs through Decomposed Subtask Instruction, KDD, 2025.
[3] GraphWiz: An Instruction-Following Language Model for Graph Computational Problems, KDD, 2024.
[4] GraphGPT: Graph Instruction Tuning for Large Language Models, SIGIR, 2024.

---

> ### Author Rebuttal · Authors · 2026-03-31
>
> **Q1.** Tool use, code execution, and RL-based training for graph reasoning have already been explored in prior work [1-3]. The main contribution seems to be the specific combination of these components, rather than a fundamentally new graph reasoning paradigm.
>
> **A1.** We respectfully clarify that GRASP is not a simple combination of components, but implements a fundamentally different approach to how LLMs interact with and reason over graph structures:
> * **Active Probing.** Unlike [1-3], which treat graphs as **static, flattened text**, GRASP enables **on-demand probing**. This bypasses the input length constraints and reasoning bottlenecks inherent in exhaustive graph flattening.
> * **Closed-loop Solving.** While prior work uses static code snippets, GRASP is the first to **interleave structural probing with real-time symbolic execution**. This allows the agent to write, execute, and debug code dynamically based on environmental feedback.
> * **Topological Grounding.** Our **two-stage RL** moves beyond semantic pattern mimicry to enforce topological reasoning. This enables handling **million-node graphs** and reasoning depths up to **60 turns**—capabilities beyond traditional static baselines.
> ---
>  **Q2.** The paper shows gains in effectiveness, but does not provide explicit analysis of runtime, token consumption, tool-call overhead, or RL training cost.
>
> **A2.** We provide detailed efficiency metrics cross representative tasks  in **Table R1**:
> - **Inference Efficiency.** GRASP significantly reduces total inference time (e.g., **5.48s vs. 13.73s** on PubMed) by offloading complex reasoning to external tools. This substantially cuts token consumption (1372 vs. 3667 on PubMed), accelerating the process as LLM latency is dominated by autoregressive decoding.
> - **Tool Overhead.** Thanks to our caching and lightweight sandbox, tool-call overhead is minimal (e.g., 0.50s out of 5.48s on PubMed).
> - **RL Training Cost:** GRASP training requires ~40 hours on 16 $\times$ H20 GPUs (800–1000 steps), which is highly cost-effective for the performance gains.
> ⠀We will incorporate these metrics and discussions into the revised manuscript.
>
> **Table R1. Efficiency Metrics (Averaged per sample).** Format: Inference time (s) / Token consumption / Tool-call time (s). Accuracy in brackets.
>
> |                             | PubMed                   | fb15k237                 | Erdős                     |
> |-----------------------------|--------------------------|--------------------------|---------------------------|
> | Qwen3-4B-Thinking (no tool) | 13.73/3667.18/-(25%)     | 22.26/5853.13/- (16%)    | 31.81/8150.06/-(38.8%)    |
> | **GRASP (Ours)**            | 5.48/1372.24/ 0.50 (90%) | 10.19/2573.47/0.22 (82%) | 9.51/2520.58/0.002(91.0%) |
> ---
>  **Q3.** Although the reported results are strong, the paper does not directly compare against several relevant recent graph reasoning methods, e.g., [1-4].
>
> **A3.** We compare **GRASP** against recent baselines: GraphAgent [1], GraphTool [2], and GraphWiz [3]. (GraphAgent is an agentic evolution of GraphGPT [4]).
> - As shown in **Table R2**, GRASP significantly outperforms recent baselines across all categories, achieving an average of **83.85%** (nearly **50% absolute gain** over the second-best 35.98%).
> - To ensure a fair comparison, we evaluate all models on **OOD benchmarks** not included in any model’s training set (**Table R3**). GRASP achieves the best average performance (**64.50%** vs. 45.83%), confirming its ability to learn intrinsic graph reasoning rather than memorizing patterns.
>
> We will add these baselines into the revised manuscript.
>
> **Table R2. Test Accuracy (%) on graph reasoning benchmarks.**
>
> |                  | Node Classification | Link Prediction | Graph QA | Graph Theory | Average   |
> |------------------|---------------------|-----------------|----------|--------------|-----------|
> | GraphAgent       | 45                  | 20.5            | 67       | 2.42         | 35.98     |
> | GraphWiz         | 1                   | 0               | 0        | 7.13         | 1.82      |
> | GraphTool        | 46                  | 17              | 54.5     | 10.33        | 34.76     |
> | **GRASP (Ours)** | 82                  | 74.5            | 91       | 89.78        | **83.85** |
>
> **Table R3. Zero-shot transferability (%) on unseen graph benchmarks.**
>
> |                  | Cora     | ml1m     | SGDS     | Average   |
> |------------------|----------|----------|----------|-----------|
> | GraphAgent       | 52.5     | 25.5     | 2.0      | 26.66     |
> | GraphWiz         | 0.0      | 18.0     | 0.0      | 6.00      |
> | GraphTool        | 55.5     | **32.0** | 50.0     | 45.83     |
> | **GRASP (Ours)** | **64.0** | 31.0     | **98.5** | **64.50** |
> -----
>
> We appreciate your time in reviewing our work and hope these responses resolve your queries. Please let us know if there are any other points you wish to discuss.

---

> > ### Author Rebuttal · Reviewer_3qBr · 2026-04-03
> >
> > Thank you for the authors' rebuttal. The authors have effectively addressed my concerns. Therefore, I will raise my score from 3 to 4. Good luck.

---

> > > ### Author Response · Authors · 2026-04-03
> > >
> > > We sincerely thank the reviewer for the positive feedback and for raising the score to Weak Accept. We are glad that our rebuttal effectively addressed your concerns. We will incorporate the efficiency metrics and extended baselines into the revised manuscript as promised.

---

### Decision · Program_Chairs · 2026-04-30

**Decision:**

Accept (regular)

**Comment:**

This paper introduces GRASP, an agentic framework for LLM-based graph reasoning that shifts from passive ingestion to active exploration via tool use. The approach integrates neighbor retrieval, code-based execution, and a two-stage RL training strategy to address key limitations such as context constraints, numerical errors, and structural hallucinations. The introduction of the RealErdos benchmark and the extensive evaluation across diverse graph tasks further strengthen the paper.

All reviewers converge on a weak accept recommendation, with consistent agreement that the paper is well-executed and empirically strong. The strengths highlighted include:

* A principled integration of components (retrieval, tool use, RL) into a coherent framework.
* Strong empirical results, including competitive or superior performance to significantly larger models and good zero-shot generalization.
* Comprehensive evaluation, spanning multiple tasks and scales, including large graph settings.

Initial concerns regarding missing baselines, ablations, efficiency, and novelty were adressed during the rebuttal phase.

While the novelty is viewed as moderate, primarily stemming from integration rather than fundamentally new primitives, the overall contribution is still considered meaningful and impactful in practice, particularly given the growing interest in tool-augmented LLM reasoning.